# The DEAD-box helicase eIF4A1/2 acts as RNA chaperone during mitotic exit enabling chromatin decondensation

Ramona Jühlen [1,2] ✉, Sabine C. Wiesmann [1,2], Anja Scheufen[1], Thilo Stausberg [1], Isabel Braun [1], Chantal Strobel[1], Carmen Llera-Brandt[1], Sabrina Rappold[1], Rabia Suluyayla[1], Marianna Tatarek-Nossol[1], Birgitt Lennartz[1], Hongqi Lue[1], Maximilian W. G. Schneider[1], Juan-Felipe Perez-Correa [1], Daniel Moreno-Andrés [1] ✉ & Wolfram Antonin [1] ✉

During mitosis, chromosomes condense and decondense to segregate faithfully and undamaged. The exact molecular mechanisms are not well understood. We identify the DEAD-box helicase eIF4A1/2 as a critical factor in this process. In a cell-free condensation assay eIF4A1/2 is crucial for this process, relying on its RNA-binding ability but not its ATPase activity. Reducing eIF4A1/2 levels in cells consistently slows down chromatin decondensation during nuclear reformation. Conversely, increasing eIF4A1/2 concentration on mitotic chromosomes accelerates their decondensation. The absence of eIF4A1/2 affects the perichromatin layer, which surrounds the chromosomes during mitosis and consists of RNA and mainly nucleolar proteins. In vitro, eIF4A1/2 acts as an RNA chaperone, dissociating biomolecular condensates of RNA and perichromatin proteins. During mitosis, the chaperone activity of eIF4A1/2 is required to regulate the composition and fluidity of the perichromatin layer, which is crucial for the dynamic reorganization of chromatin as cells exit mitosis.

During mitosis, chromatin undergoes dramatic structural and functional changes, allowing it to segregate faithfully into daughter cells. Nuclear chromatin condenses into individualized rod-shaped chromosomes, which are then captured and positioned at the metaphase plate by the spindle apparatus. During the transition from metaphase to anaphase, sister chromatids are segregated into the two emerging daughter cells. In late anaphase and telophase, the chromatids decondense and merge in each forming cell, typically into one large chromatin mass around which a nuclear envelope forms, giving rise to a new nucleus.

While the cell cycle regulatory network driving mitotic entry and progression[1] and changes in microtubule dynamics and organization for spindle function[2] have been intensively studied, changes in chromatin structure during mitosis remain less well defined[3,4]. Mitotic chromatin compaction is thought to be driven by a combination of DNA loop extrusion[5], mediated by condensins[6], and chromatin soluble phase transition[7,8], controlled by histone modifications[9,10]. A perichromosomal layer covers the mitotic chromosomes except for the centromeric regions[11,12]. Several nuclear and nucleolar proteins and RNAs, such as snRNAs, form this layer. A key component and organizer is Ki-67[13], which may enable chromosome individualization and independent movement through its surfactant-like properties[14], but alternative roles, such as in the organization of mitotic chromosomes[15], have been suggested. With the onset of anaphase and chromatid segregation, mitotic chromatin decondenses in a poorly understood manner. Repo-Man recruits the protein phosphatase PP1γ to chromatin during this process, counteracting phosphorylation events critical during early mitosis[16,17]. The ATPase p97 extracts the mitotic kinase

[1]Institute of Biochemistry and Molecular Cell Biology, Medical School, RWTH Aachen University, Aachen, Germany. [2]These authors contributed equally: Ramona Jühlen, Sabine C. Wiesmann. ✉e-mail: rjuehlen@ukaachen.de; dmoreno@ukaachen.de; wantonin@ukaachen.de

Aurora B from chromatin to allow nuclear reassembly[18], and another ATPase complex formed by RuvBL1 and RuvBL2 is required for chromatin decondensation during mitotic exit[19]. In this process, Ki-67 switches its properties, controlled by massive dephosphorylation, to enable the coalescence of segregating sister chromatids into single chromosome masses for nuclear reformation[20,21].

Here, we show that the DEAD box-helicase eIF4A1/2 plays a crucial role during mitotic exit. Although best known for its function as a translation initiation factor as part of the eIF4F complex, eIF4A1/2 acts outside of this complex and independently from translation during mitotic exit. Instead, it plays crucial functions as an RNA chaperone, balancing the RNA content of the perichromatin layer and critically enabling chromatin decondensation during telophase.

## Results

### eIF4A1/2 is required for chromatin decondensation

We have previously developed a cell-free assay using *Xenopus* egg extracts and mitotic chromatin clusters to study chromatin decondensation[19]. Chromatin decondensation can be followed and accurately quantified by measuring the border smoothness of the chromatin area at the end of the reaction (Fig. 1). While searching for new factors that affect chromatin condensation, we discovered that immunodepletion of the translation initiation factor eIF4A1/2 (also known as DDX2A and DDX2B, respectively) blocks chromatin decondensation (Fig. 1A–C), quantifiable as a reduction in average border smoothness from 0.94 to 0.72. The antibodies used for immunodepletion were raised against full-length *Xenopus* eIF4A1. Because of the high sequence identity between eIF4A1 and eIF4A2 (91 % amino acid sequence identity in *Xenopus laevis*), which both act as translation initiation factors[22,23], both proteins are likely recognized by the antibodies and immunodepleted. Adding recombinant eIF4A1 or eIF4A2 to eIF4A1/2 depleted extracts to approximately 6 µM, which equals the endogenous concentration of eIF4A1 in egg extracts (Fig. 1C), restored chromatin decondensation, demonstrating the specificity of the depletion. However, adding eIF4A3, also known as DDX48, a splicing and nuclear export cofactor[22,24,25], which shows only 66 or 67 % sequence identity to eIF4A1 or eIF4A2, respectively, did not restore chromatin decondensation.

eIF4A1/2 belongs to the DEAD-box helicase family and catalyzes the ATP-dependent unwinding of RNA structures[26]. The helicase core is formed by two RecA-like domains essential for ATP binding and hydrolysis, RNA binding, and unwinding activity. These different functions can be separated with mutants in the RecA-like domains: While an R362/365Q exchange blocks RNA-binding, an E183Q mutation blocks the protein's ATPase activity[27,28]. In the cell-free decondensation assay, the addition of the eIF4A1 R362/365Q mutant did not restore chromatin decondensation in eIF4A1/2 depleted extracts, indicating that RNA binding activity is crucial for eIF4A1 function in this process (Fig. 1D–F). Interestingly, at least in this system, the E183Q mutant can restore chromatin decondensation.

Given that eIF4A1's RNA-binding function is essential for chromatin decondensation in vitro, we investigated whether an excess of RNA could hinder this process. Our findings revealed that adding exogenous RNA effectively blocked chromatin decondensation (Fig. 1G, H), alleviated by the co-addition of RNase. We hypothesize that this inhibitory effect arises from the saturation of endogenous eIF4A1 activity, which could be, accordingly, counteracted by providing an excess of recombinant eIF4A1.

Next, we aimed to confirm the potential function of eIF4A1/2 in chromatin decondensation in cells. For this, we monitored mitotic progression through live-cell imaging in HeLa cells stably expressing H2B-mCherry after siRNA-mediated downregulation of eIF4A1, the main eIF4A1/2 orthologue in these cells[29], using three different oligos. Analysis with CellCognition[30] revealed that eIF4A1 downregulation extended telophase to 41, 45, and 40 min compared to control cells

(25 minutes, Fig. 2A). Downregulation of PP2A, a phosphatase controlling mitotic exit, served as a positive control and extended prometaphase, anaphase and telophase as expected[30], whereas eIF4A1 downregulation did not consistently prolong other mitotic phases (Fig. 2A and Supplementary Fig. 1A–C). Downregulation of the paralogous eIF4A2 and eIF4A3 using three different oligos did not extend telophase (Supplementary Fig. 1D and I), indicating that the effect is specific to eIF4A1.

Despite efficient downregulation of eIF4A1 by all three oligos, we observed increased eIF4A2 expression levels, as detected by Western blotting with antibodies recognizing both eIF4A1 and eIF4A2 (Supplementary Fig. 1H). Treatment with oligos against both eIF4A1 and eIF4A2 efficiently decreased the eIF4A1/2 expression (Fig. 2D) and further extended telophase to 64 min (Fig. 2A). Analyzing chromatin area increase as a proxy for chromatin decondensation using LiveCellMiner[31], we found that eIF4A1 downregulation slightly reduced decondensation kinetics after metaphase to anaphase transition, with more pronounced effects when both eIF4A1 and eIF4A2 were downregulated (Fig. 2B). Downregulation of the paralogous eIF4A2 and eIF4A3 using three different oligos did not reduce decondensation kinetics (Supplementary Fig. 1E).

Downregulation of the expression of other proteins that function with eIF4A1 in the eIF4F complex during translation initiation did not extend telophase or affect chromatin decondensation except for eIF4H, which showed an increase in telophase length and decrease in chromatin decondensation (Supplementary Fig. 1F, G). Given that protein synthesis is inactive in the cell-free assay[19] and translation inhibitors like cycloheximide or harringtonine do not mimic the chromatin decondensation defect of eIF4A1/2 downregulation (Supplementary Fig. 2A and B), these data suggest that eIF4A1/2 impacts mitotic chromatin decondensation independently of translation. Reportedly, cellular levels of eIF4A1/2 far exceed those of other eIF4F complex members[32,33], and eIF4A1/2 downregulation does not significantly affect their expression levels (Supplementary Fig. 1J). Therefore, the observed phenotype upon eIF4A1/2 downregulation is likely not mediated through the eIF4F complex and protein translation.

Although eIF4A1/2 downregulation slowed down chromatin decondensation in cells, the chromatin area of interphase nuclei remained largely unaffected, with an average of 236 µm² upon eIF4A1/2 downregulation versus 255 µm² in the control (Fig. 2C). High-resolution live-cell imaging during mitotic exit further showed that eIF4A1/2 downregulation significantly reduced the rate of chromatin area increase during telophase (Fig. 2E, F) without an increase in detectable mitotic errors like chromatin bridges or lagging chromosomes. The effect of eIF4A1/2 downregulation is specific to chromatin decondensation, as nuclear import re-initiation is not affected by eIF4A1/2 siRNA treatment in a HeLa cell line stably expressing H2B-mCherry and the nuclear import marker IBB-EGFP (the importin β binding domain of importin α fused to EGFP[34]) (Supplementary Fig. 3A, B).

To evaluate the specificity of the siRNA treatment against eIF4A1/2, we conducted an analysis of chromatin decondensation in HeLa cells transfected with either EGFP-eIF4A1 or a version of the construct that is resistant to siRNA (Fig. 2G and H). Our findings revealed that while the siRNA treatment effectively reduced the expression levels of endogenous eIF4A1/2 and EGFP-eIF4A1, the siRNA-resistant variant of EGFP-eIF4A1 remained unaffected. Notably, cells expressing this siRNA-resistant version exhibited decondensation kinetics upon the downregulation of endogenous eIF4A1/2 that were comparable to those observed in control-treated cells. In contrast, cells harboring the siRNA-sensitive EGFP-eIF4A1 construct demonstrated a significant reduction in decondensation kinetics.

These data indicate that eIF4A1/2 is required for efficient chromatin decondensation during mitotic exit. Its depletion blocks

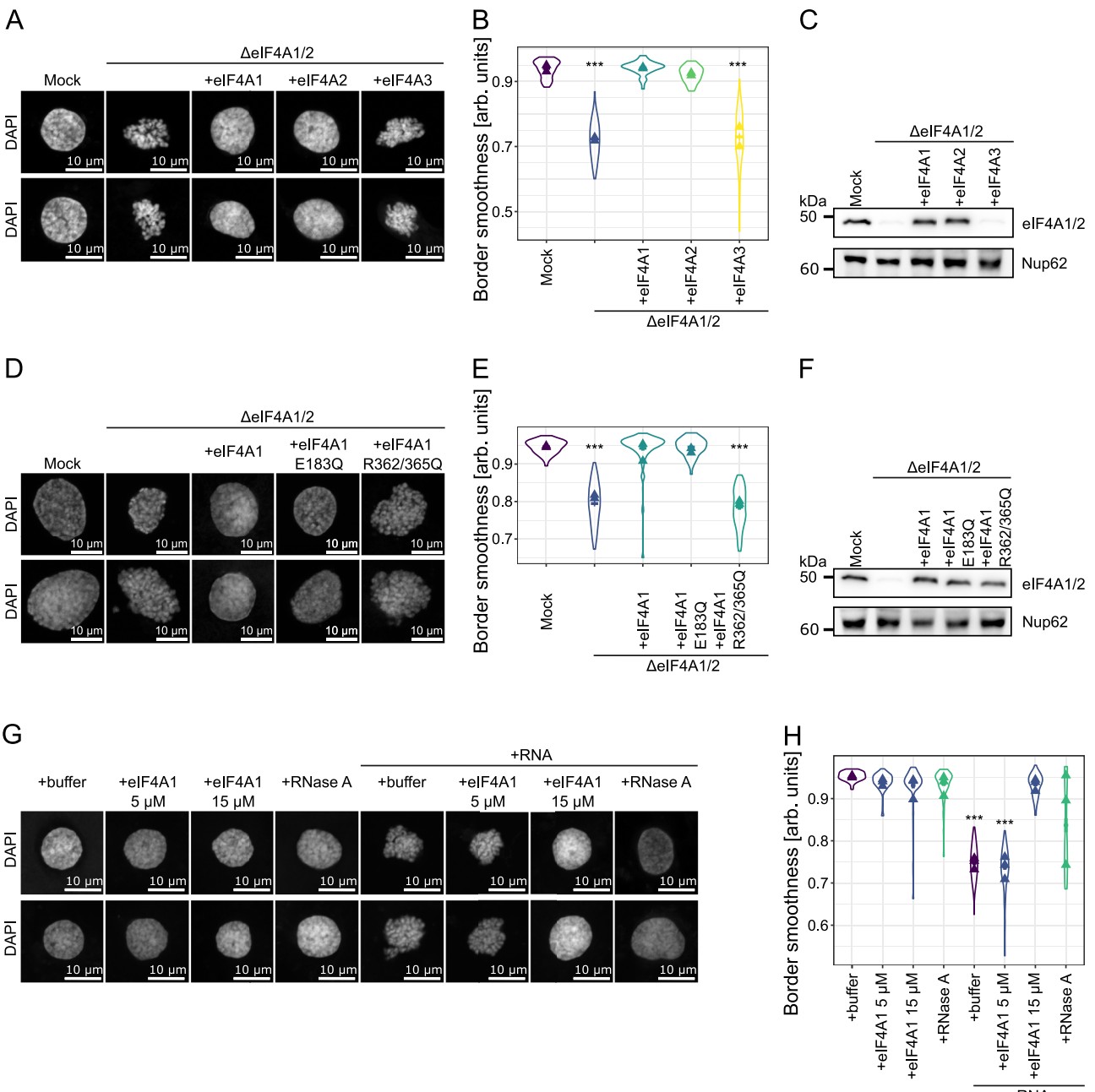

**Fig. 1 | eIF4A1/2 is important for in vitro chromatin decondensation. A** Mitotic chromatin clusters were decondensed in control- (Mock) and eIF4A1/2-depleted (ΔeIF4A1/2) *Xenopus* egg extracts supplemented with buffer or recombinant *Xenopus* eIF4A1, eIF4A2 or eIF4A3. Samples were fixed after 2 h with 4% PFA and 0.5% glutaraldehyde, stained with DAPI, and analyzed by confocal microscopy. **B** For decondensation quantification, the boundary smoothness of the chromatin was analyzed. Means of two independent experiments (triangles), overall mean ± s.e.m. of in total more than 50 chromatin substrates for each condition as in (**A**). At some points, errors might be too small to be visible. Two-tailed unpaired Mann-Whitney *U* test (ΔeIF4A1/2, ***$P = 3.3 \times 10^{-18}$; +eIF4A3, ***$P = 2.63 \times 10^{-18}$). **C** Western blot analysis with a *Xenopus* eIF4A1/2 antibody of Mock- and eIF4A1/2-depleted extracts, without or with the addition of recombinant *Xenopus* eIF4A1, eIF4A2 or eIF4A3. Nup62 is used as a loading control. **D** Mitotic chromatin clusters were decondensed and analyzed as in (**A**) but supplemented with buffer or recombinant wild-type eIF4A1, the ATPase deficient E183Q, or the RNA-binding deficient R362/365Q mutant. **E** The violin plot of the chromatin border smoothness. Means of

three independent experiments (triangles), overall mean ± s.e.m. of in total more than 50 chromatin substrates for each condition as in (**D**). At some points, errors might be too small to be visible. Two-tailed unpaired Mann-Whitney *U* test (ΔeIF4A1/2, ***$P = 7.65 \times 10^{-19}$; + eIF4A1 R362/365Q, ***$P = 4.96 \times 10^{-19}$). **F** Western blot analysis with a *Xenopus* eIF4A1/2 antibody of Mock-, eIF4A1/2-depleted, or eIF4A1/2-depleted extracts supplemented with the indicated eIF4A1 proteins. Nup62 is used as a loading control. **G** Mitotic chromatin clusters were decondensed in *Xenopus* egg extracts in the absence or presence of 0.05 mg/ml RNA supplemented with buffer, 5 or 15 μM recombinant *Xenopus* eIF4A1 or 50 μg/ml RNase A and analyzed as in (**A**). **H** Violin plot shows the means of three independent experiments (triangles) and the overall mean ± s.e.m., of in total more than 40 chromatin substrates for each condition as in (**E**). At some points, errors might be too small to be visible. Two-tailed unpaired Mann-Whitney *U* test ( + RNA, ***$P = 1.63 \times 10^{-18}$; + RNA + eIF4A1 5 μM, ***$P = 1.63 \times 10^{-18}$). Source data are provided as a Source Data file. Scale bars: 10 μm.

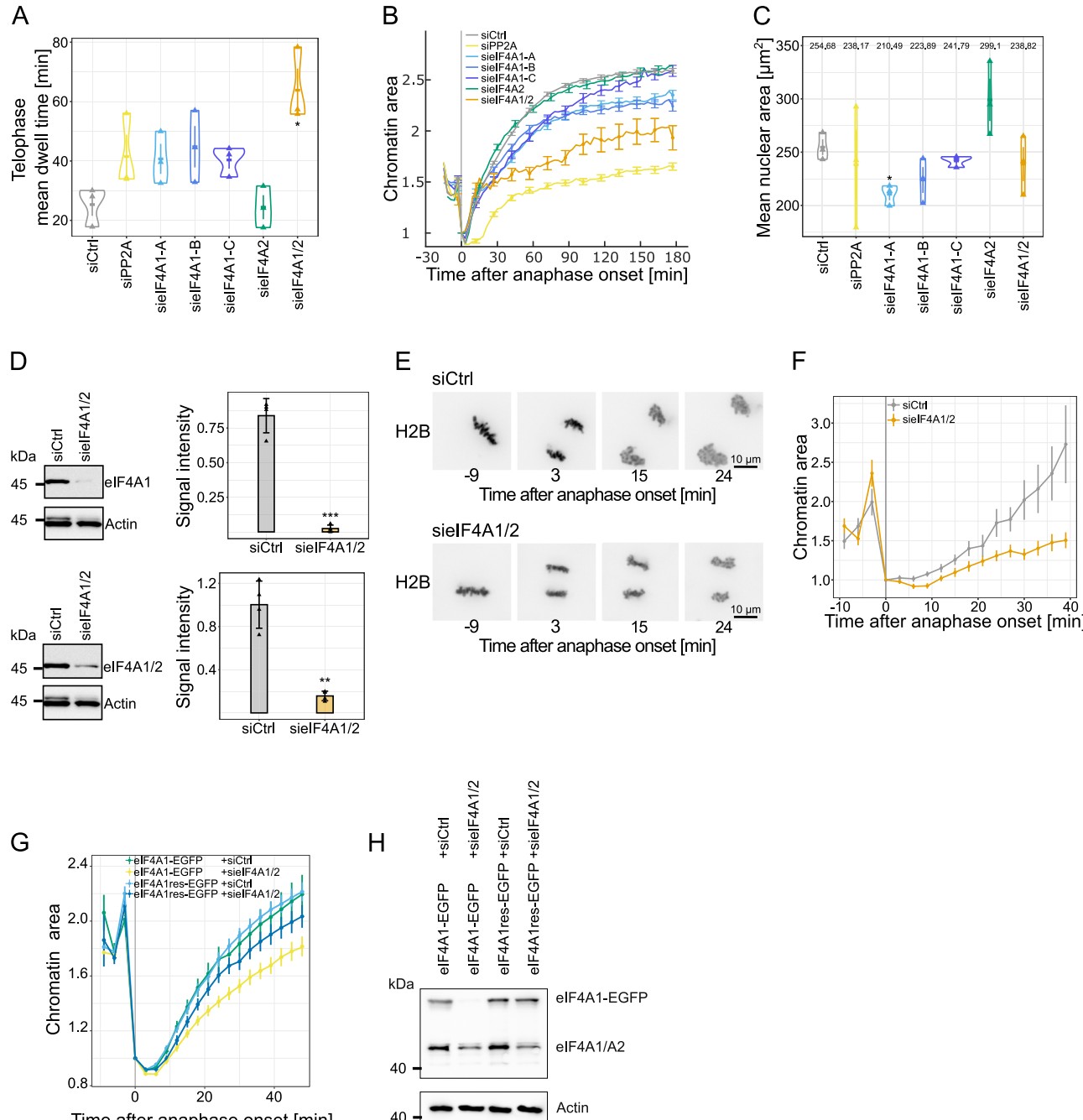

**Fig. 2 | Downregulation of eIF4A1/2 decelerates mitotic chromatin decondensation in cells. A** Telophase duration in H2B-mCherry HeLa cells, transfected with 20 nM siRNA control or against PP2A, eIF4A1 (three different oligos), eIF4A2 or a combination of eIF4A1 (oligo A) and eIF4A2, analyzed by live cell imaging 48–63 h post-transfection. Mean telophase time of at least 150 mitotic events per condition from three independent experiments with the individual (triangles) and overall means ± s.e.m. Two-tailed unpaired *t* test with Welch's correction (sieIF4A1/2, *P = 0.02). **B** Chromatin area analysis from (**A**) with at least 90 mitotic events per condition, normalized to the first anaphase frame. Lines represent means ± s.e.m. **C** Nuclear area of interphase cells, defined by 10 frames before interphase-prophase transition, from (**A**), 90 cells per condition. Means of three independent experiments (triangles) and overall mean ± s.e.m. with values for each transfection. Two-tailed unpaired *t* test with Welch's correction (sieIF4A1-A, *P = 0.01). **D** eIF4A1/2 Western blot of H2B-mCherry HeLa cells, 72 h post-transfection with 20 nM siRNA oligos, corresponding to (**A**–**C**), with eIF4A1 antibodies (upper) or antibodies recognizing eIF4A1 and eIF4A2 (lower panel). Actin serves as loading control. Signal intensity, normalized to actin, of four independent experiments (triangles) and overall mean ± sd. Two-tailed unpaired *t* test with Welch's correction (eIF4A1, ***P = 6.73 × 10⁻⁴; eIF4A1/2, **P = 3.58 × 10⁻³). **E** Time-lapse images of H2B-mCherry HeLa cells, transfected with 20 nM control or a combination of eIF4A1 and eIF4A2 siRNA oligos for 72 h, normalized to the first anaphase frame. **F** Analysis of the experiments as in (**E**). Dots represent mean ± s.e.m. from 11 (siCtrl) and 10 (sieIF4A1/2) cells per condition. **G** Chromatin area quantification of H2B-mCherry HeLa cells, co-transfected with eIF4A1-EGFP or siRNA resistant eIF4A1-EGFP and 40 nM control or a combination of eIF4A1 and eIF4A2 siRNA oligos for 48 h, normalized to the first anaphase frame. Dots represent mean ± s.e.m. from each condition with 47 (eIF4A1-EGFP+ siCtrl), 57 (eIF4A1-EGFP + sieIF4A1/2), 72 (eIF4A1res-EGFP + siCtrl) or 43 (eIF4A1res-EGFP + sieIF4A1/2) daughter chromatin masses. **H** eIF4A1/2 western blot of H2B-mCherry HeLa cells, 48 h post-transfection with 40 nM siRNA oligos and eIF4A1-EGFP or eIF4A1res-EGFP. Actin serves as loading control. Source data are provided as a Source Data file. Scale bars: 10 μm.

chromatin decondensation in *Xenopus* egg extracts, and down-regulation of its expression slows down chromatin decondensation in HeLa cells and accordingly increases the time the cells are assigned in telophase. Like HeLa cells, RPE cells stably expressing H2B-RFP as a chromatin marker show a longer dwell time in telophase upon eIF4A1/2 downregulation (Supplementary Fig. 3C–E).

## eIF4A1/2 downregulation affects the perichromatin layer

Mitotic chromosomes are covered by perichromatin[11,12,35], a layer of RNA and proteins essential for individualizing chromosomes. We wondered whether eIF4A1/2 acts as an RNA-helicase on the perichromatin, so we imaged RNA on the surface of mitotic chromosomes. To do this, we labeled the newly synthesized RNA with a 5-ethynyluridine pulse three hours before the synchronized cells entered mitosis and fixation, following a recently established protocol[21], and imaged cells during the different mitotic phases. While the average fluorescence of RNA remained stable during metaphase and anaphase, there was a slight increase of the signal in telophase in control-treated cells. Notably, the downregulation of eIF4A1/2 led to a reduction in RNA signal across all three phases (Fig. 3A, B).

Several proteins in the perichromatin layer can bind RNA such as DDX18 or DDX27[36,37], and we investigated whether their levels are influenced by the decreased RNA amount on mitotic chromosomes following eIF4A1/2 downregulation. Immunofluorescence using DDX18 and DDX27 antibodies on fixed HeLa cells expressing H2B-mCherry revealed distinct labeling of chromosomes by DDX18 and DDX27, with the highest intensity observed on anaphase chromosomes (Fig. 3C, D, G, H). After downregulating eIF4A1/2, the perichromatin signals for both DDX18 and DDX27 were significantly diminished during metaphase, anaphase, and telophase (Fig. 3D, H). This was accompanied by the appearance of cytosolic DDX18 and DDX27 spots (Fig. 3E, F, I, J).

As DDX18 and DDX27 are as RNA helicase direct interactors of RNA, we aimed to expand the analysis towards perichromatin markers which do not possess a reported RNA binding activity. eIF4A1/2 downregulation in a HeLa cell line stably expressing H2B-mPlum and the perichromatin marker EGFP-CCDC137[37] revealed a mislocalization of CCDC137 away from the chromosomal periphery into clusters (Fig. 4A–D and Supplementary Fig. 4A). This phenotype is also as reportedly[13,20,35] observed, albeit more penetrant, upon downregulation of Ki-67, a central organizer of the perichromatin[13,35,36].

HeLa cells stably expressing EGFP-Ki-67 showed upon eIF4A1/2 downregulation sticky chromosomes after early anaphase resembling anaphase bridges during sister chromatid separation and/or chromatin bridges during daughter cell formation (Fig. 4E and Supplementary Fig. 4B–D). Chromatin localization of EGFP-Ki-67 was unchanged in early mitosis but slightly decreased in later mitotic stages (Supplementary Fig. 4E–G). Consistently, immunofluorescence with a Ki-67 antibody on fixed HeLa cells expressing H2B-mCherry revealed a reduced Ki-67 chromatin signal in anaphase and early telophase (Fig. 4F, G). At the same time, no difference was detected in interphase, where Ki-67 decorates the surface of nucleoli, in earlier mitotic stages or in late telophase (Supplementary Fig. 4H–K). Nucleolar size and number of nucleoli per nucleus were unaltered in these cells after eIF4A1/2 downregulation (Supplementary Fig. 4L, M), which suggests a role of eiF4A1/2 in the control of the dynamics of the perinuclear layer during late mitosis but not in nucleolar reformation during late telophase and G1. Together, these data indicate that the downregulation of eIF4A1/2 affects the perichromatin layer, especially in the second half of mitosis.

## eIF4A1 tethering to mitotic chromosomes accelerates chromatin decondensation

In interphase, eIF4A1 and eIF4A2 are localized in the cytoplasm while eIF4A3 is, as a spliceosome factor, nuclear ([24,38], Fig. 5).

Immunostaining revealed that during mitosis, eIF4A1/2 is distributed throughout the cytoplasm but excluded from mitotic chromosomes (Fig. 5A and B). Similarly, eIF4A1-EGFP is excluded from mitotic chromosomes and localized cytoplasmic in interphase (Fig. 5C–E) both in fixed samples and during live cell imaging. This localization pattern was unchanged for the ATPase-deficient E183Q and the RNA-binding deficient mutant R362Q (Fig. 5E). Using live cell imaging of EGFP-tagged proteins of the eIF4F complex directly interacting with eIF4A1/2, we could see the same exclusion from mitotic chromatin (Fig. 5E and Supplementary Fig. 5), and this pattern was also unchanged for the eIF4A1 E183Q and R362Q mutant (Supplementary Fig. 5). At later stages of mitotic exit, eIF4A1 and eIF4A2 are quickly excluded from the reforming nucleus as compared, for example, to free EGFP. We thus asked whether forcing eIF4A1 into the nucleus with a nuclear localization signal (NLS) would enhance chromatin decondensation. Expression of eIF4A1-EGFP-NLS slightly enhanced the decondensation rate, especially during the late telophase stages, compared to eIF4A1-EGFP (Supplementary Fig. 6). Interestingly, despite an NLS signal, eIF4A1-EGFP-NLS was also quickly excluded from the reforming nucleus (Supplementary Fig. 6A) and only slowly accumulated in the nucleus in the following G1 within the next two hours. Also, in this context, the eIF4A1 E183Q and R362Q mutant did not result in changes.

We next aimed to enhance eIF4A1 concentration on mitotic chromosomes. For this, we used the rapamycin-dependent interaction between FRB and FKBP by transfecting EGFP-FRB-eIF4A1 into HeLa cells stably expressing H2B-mPlum-FKBP[39]. Here, EGFP-FRB-eIF4A1 can be tethered to mitotic chromosomes quickly and efficiently by rapamycin addition and the fate of the chromatin followed by high-resolution live cell imaging for the remaining mitosis. Chromatin tethering of eIF4A1 in metaphase enhanced chromatin decondensation after metaphase to anaphase transition compared to the control (Fig. 6A and B). In contrast, both the E183Q (Fig. 6C and D) and the R362Q mutant (Fig. 6E and F) did not affect chromatin decondensation upon tethering, indicating that, in this experimental set-up, the ATPase activity as well as RNA-binding ability is required. Also, metaphase chromatin tethering of eIF4A2 and, surprisingly, eIF4A3 enhanced chromatin decondensation (Fig. 6G–J). No effect on chromatin decondensation rates is seen upon tethering of eIF4B, G1, or H known to enhance the catalytic helicase activity of eIF4A1 (Supplementary Fig. 7A–F). Consistently, tethering of eIF4A1 to metaphase chromatin does not co-recruit either of these co-factors (Supplementary Fig. 7G). Thus, the enhanced decondensation rate observed upon eIF4A1/2 tethering to chromatin does not require the other members of the eIF4F complex directly interacting with eIF4A1/2.

Considering that the downregulation of eIF4A1/2 diminished the labeling intensity of perichromatin markers, we determined whether eIF4A1 chromatin tethering affects Ki-67 or DDX18 staining intensity on perichromatin, which, however, was largely unaffected (Supplementary Fig. 6E–H).

Given that chromatin tethering of eIF4A3 slightly enhances chromatin decondensation, we tested whether eIF4A3 overexpression could rescue, at least in part, the effect of eIF4A1/2 downregulation. Indeed, overexpression of EGFP-eIF4A3 reduces the impact of eIF4A1/2 downregulation compared to the EGFP control (Fig. 6K, L).

## eIF4A1 resolves RNA-helicase condensates

Our data suggest that eIF4A1 and eIF4A2 do not localize to the surface of mitotic chromosomes; rather, they play a role in forming or maintaining the perichromatin layer, which encompasses its protein and RNA constituents. Recent studies have highlighted DEAD-box helicases as key regulators of phase separation[40], with the perichromatin being proposed as a phase-separated compartment of RNA and proteins[12,41]. The interplay of multivalent and transient interactions sustains RNA, protein, or RNA-protein assemblies in a dynamic and reversible state -

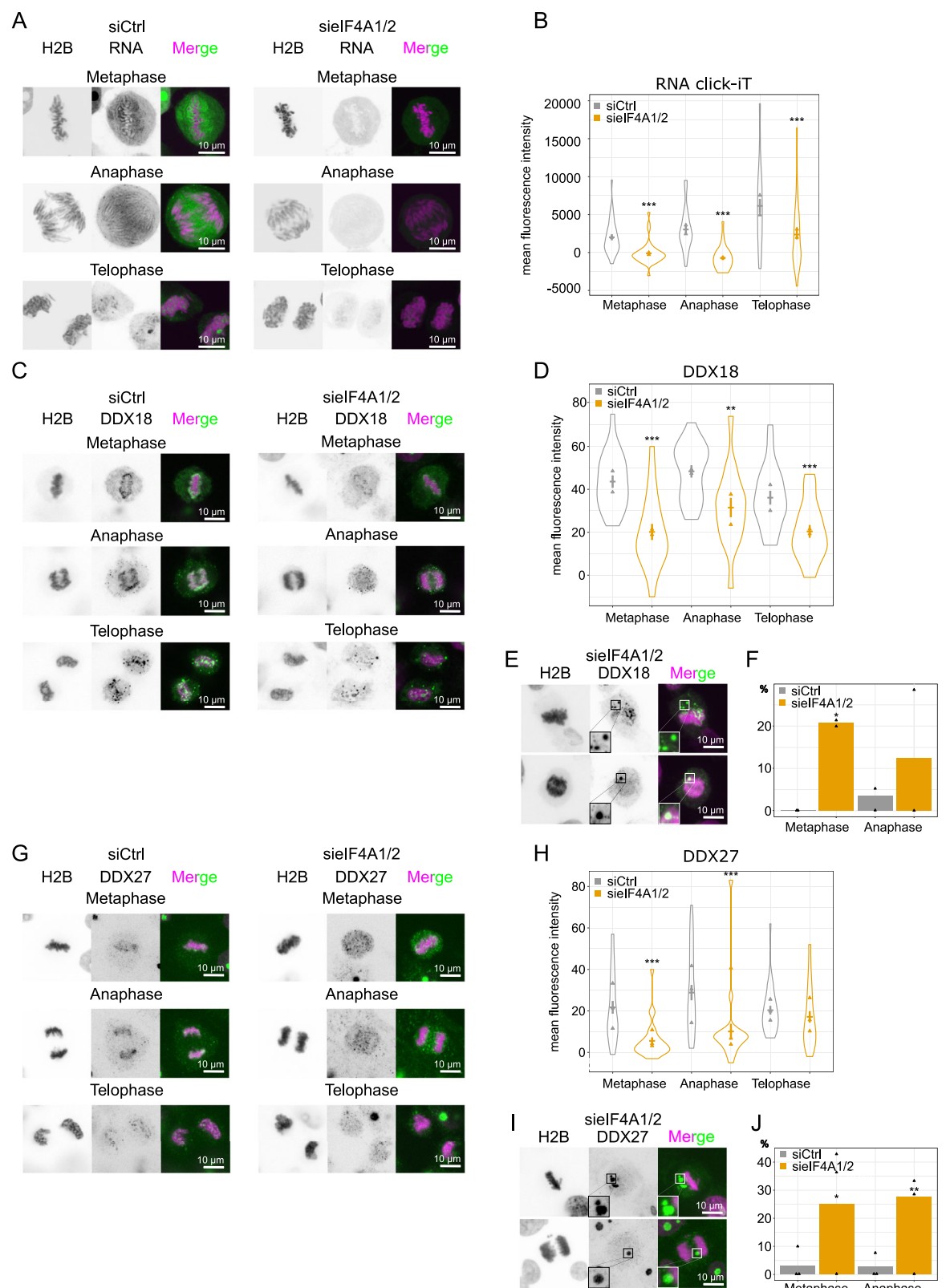

characterized by biomolecular condensation or liquid-liquid phase separation. We, therefore, hypothesized that eIF4A1/2 may influence these phase-separated compartment on the surfaces of mitotic chromosomes, envisioning two potential scenarios: In the first scenario, eIF4A1/2 might directly bind to and stabilize the perichromatin layer. However, we consider this unlikely since eIF4A1/2 does not appear to enrich on the surface of mitotic chromosomes; it is, however, possible

that their high cytosolic concentrations obscure any potential enrichment. The second scenario posits that cytosolic eIF4A1/2 rather facilitates RNA mobility, thereby promoting dynamic exchanges between cytosolic RNA and the perichromatin RNA pool.

To differentiate between these scenarios, we conducted in vitro experiments to form RNA-protein droplets (Fig. 7A) utilizing three helicases associated with the perichromatin layer: DDX18, DDX21,

**Fig. 3 | Downregulation of eIF4A1/2 affects the RNA content and RNA helicases of the perichromosomal layer. A** High-resolution imaging of EU-labeled RNA (green in overlay) in fixed H2B-mCherry HeLa cells (magenta in overlay), transfected with 20 nM control or a combination of eIF4A1 and eIF4A2 siRNA oligos for 72 h. **B** RNA mean fluorescence intensity on the chromatin in metaphase, anaphase, and telophase. The violin plots show the means of two independent experiments (triangles) each including at least 26 cells per condition and the overall mean ± sd. Two-tailed unpaired Mann-Whitney $U$ test (metaphase, ***$P = 1.3 \times 10^{-6}$; anaphase, ***$P = 3.8 \times 10^{-7}$; telophase, ***$P = 9.6 \times 10^{-4}$). **C** DDX18 immunostaining (green) of H2B-mCherry HeLa cells (magenta), transfected with 20 nM control or a combination of eIF4A1 and eIF4A2 siRNA oligos for 72 h. **D** DDX18 mean fluorescence intensity on the chromatin in metaphase, anaphase, and telophase as in (**C**). The violin plots show the means of two independent experiments (triangles) each including at least 7 cells per condition and the overall mean ± s.e.m. Two-tailed unpaired $t$ test with Welch's correction (metaphase, ***$P = 1.7 \times 10^{-5}$; anaphase,

**$P = 4.0 \times 10^{-3}$; telophase, ***$P = 9.4 \times 10^{-4}$). **E** Example images of cytoplasmic DDX18 foci in the eIF4A1/2 knockdown in metaphase and anaphase. **F** Percentage of cells showing big cytoplasmic DDX18 foci as in (**C**). Means of two independent experiments (triangles). Two-sided Fisher's exact test (metaphase, *$P = 1.11 \times 10^{-2}$). **G** DDX27 immunostaining (green) of H2B-mCherry HeLa cells (magenta) transfected with 20 nM control or a combination of eIF4A1 and eIF4A2 siRNA oligos for 72 h. **H** Quantitation of the DDX27 mean fluorescence intensity as in (**D**). The violin plots show the means of three independent experiments (triangles) each including at least 3 cells per condition and the overall mean ± s.e.m. Two-tailed unpaired Mann-Whitney $U$ test (metaphase, ***$P = 6.6 \times 10^{-6}$; anaphase, ***$P = 9.5 \times 10^{-5}$; telophase, ns $P = 0.06$). **I** Example images of cytoplasmic DDX27 foci in the eIF4A1/2 knockdown in metaphase and anaphase. **J** Percentage of cells showing big cytoplasmic DDX27 foci. Means of three independent experiments (triangles). Two-sided Fisher's exact test (metaphase, *$P = 1.63 \times 10^{-2}$, anaphase, **$P = 8.1 \times 10^{-3}$). Source data are provided as a Source Data file. Scale bars: 10 μm.

and DDX27[36,37,42]. Fluorescently labeled RNA confirmed that these droplets contained both RNA and the respective helicases (Supplementary Fig. 8A). In contrast, under our experimental conditions, neither eIF4A1 nor eIF4A2 - or even eIF4A3 - could form RNA-protein condensates (Supplementary Fig. 8B), aligning with previous findings regarding eIF4A1[40]. To test whether eIF4A1 could enrich in DDX18, DDX21, and DDX27 droplets we added mCherry-eIF4A1 to the respective samples. While mCherry-eIF4A1 did not localize within droplets (Fig. 7B), we observed that their presence caused DDX18, DDX21, and DDX27 droplets to occur less frequent. In contrast, mCherry-DDX27 enriched in DDX18- or DDX21-RNA condensates (Fig. 7C).

To further investigate the interplay of eIF4A-helicases and condensates, we titrated increasing amounts of untagged eIF4A1 into DDX18, DDX21, and DDX27 droplets and noted their dissolution (Fig. 7D and Supplementary Fig. 8C–E). Similarly, rising concentrations of eIF4A2 and eIF4A3 resulted in the dissolution of RNA-DDX18 droplets; however, for eIF4A3 higher concentrations were necessary (Fig. 7E and Supplementary Fig. 8F, G). Notably, the ATPase-deficient E183Q mutant of eIF4A1 could still resolve RNA-DDX18 droplets (Fig. 7F and Supplementary Fig. 8H), whereas the R362/365Q mutant - deficient in RNA binding - failed to do so (Fig. 7F and Supplementary Fig. 8I). This indicates that while ATPase activity is not essential for this process, binding to RNA is crucial for droplet resolution.

The data presented suggest that eIF4A helicases can resolve biomolecular condensates formed by RNA-protein interactions. Consequently, we propose that eIF4A1 does not merely function as a binding factor for perichromatin; rather, it facilitates RNA mobility by acting as a chaperone, which is essential for its role in chromatin decondensation. Disrupting the RNA-binding capability of eIF4A1 impedes this process, aligning with the observation that the R362/365Q mutant fails to support chromatin decondensation both in vitro (as shown in Fig. 1) and in cellular contexts (as illustrated in Fig. 6).

## Discussion

Here, we show that eIF4A1/2 has a so far unappreciated function in mitosis, independent of its canonical role in translational initiation as part of the eIF4F complex. Its depletion in the cell-free system or downregulation in cells blocks or slows down chromatin decondensation during mitotic exit. Vice versa, enhancing eIF4A1/2 concentration on mitotic chromatin augments chromatin decondensation.

eIF4A1 is best known for its crucial role in initiating cap-dependent translation in eukaryotes[43]. eIF4A1 is an essential part of the eIF4F cap-binding complex, which recruits the 40S ribosomal subunit to the 5′ cap of mRNA. Here, eIF4A1 is involved in loading the mRNA onto the 43S pre-initiation complex. Then, eIF4A1 functions as an ATP-dependent RNA helicase, unwinding stable secondary structures in the 5′-untranslated region of mRNA during the scanning

process, as ribosomes have a weak capacity to unwind mRNA secondary structures on their own. While eIF4A1 is a weak helicase, its efficiency is significantly improved when it forms complexes with cofactor proteins such as eIF4G, eIF4B, and eIF4H. The orthologue eIF4A2 is thought to have broadly similar functions as eIF4A1 in translation initiation[22], but with eIF4A1 being the primary player. Accordingly, eIF4A1 but not eIF4A2 is essential in mice[44] and cellular systems[45]. Our data indicate that the eIF4A1/2 function during mitotic exit is independent of translation. The *Xenopus* egg extracts used for the in vitro chromatin decondensation contain cycloheximide to prevent protein synthesis, especially of cyclin B, to maintain the cell cycle state[19,46]. Similarly, translation inhibitors like harringtonine or cycloheximide do not inhibit chromatin decondensation in cells (Supplementary Fig. 2). Consistently, the other components of the eIF4F complex, which play important roles in translation initiation, do not function in chromatin decondensation (Supplementary Fig. 1, 7).

What, instead, could be a function of an RNA helicase on chromatin decondensation? Mitotic chromosomes are covered by a layer of proteins and RNAs known as perichromatin or chromosomes' periphery compartment[11,12], which might comprise one-third of the metaphase chromosome volume[47]. It has been speculated that the perichromatin forms a liquid-like layer on chromosomes[12,41,48]. The highly abundant perichromatin protein Ki-67, because of its structural disorder, presence of low complexity regions, and ability for multivalent interactions, is prone to phase separation, which might be especially suited for this[48]. In this context, Ki-67 recruits ribosomal RNA to the perichromatin[49]. Similarly, RNA and RNA-binding proteins, including several helicases, are components of biomolecular condensates because of the intrinsic multivalency of the interactions[50]. Notably, the perichromatin components and helicases DDX18, DDX21, and DDX27 form biomolecular condensates with RNAs in vitro (Fig. 7 and ref. 51). In contrast, eIF4A1/2 and eIF4A3 do not form such biomolecular condensates with RNA (Supplementary Fig. 8B), as previously reported for eIF4A1[40]. Our data further indicate that eIF4A1/2 does not enrich within the DDX18-, DDX21-, or DDX27-RNA condensates (Fig. 7B) or within the perichromatin itself (Fig. 5). Instead, eIF4A1/2 can resolve pure RNA condensates[52] and, as demonstrated here, can also disassemble condensates formed by RNA in conjunction with DDX18, DDX21, or DDX27 (Fig. 7D, E). We propose that the action of eIF4A1/2 is localized within the mitotic cytosol and functions as a chaperone activity that mobilizes RNAs - preventing their entrapment in cytosolic RNA droplets - and facilitates their participation in the perichromatin layer (Fig. 7G). Consequently, downregulation of eIF4A1/2 expression results in a diminished detectable RNA signal on mitotic chromatin (Fig. 3), likely leading to a corresponding reduction in the proteins typically associated with the perichromatin layer (Figs. 3 and 4). Ki-67 has been proposed to facilitate the merging of segregating sister chromatids into a unified chromosome mass, which

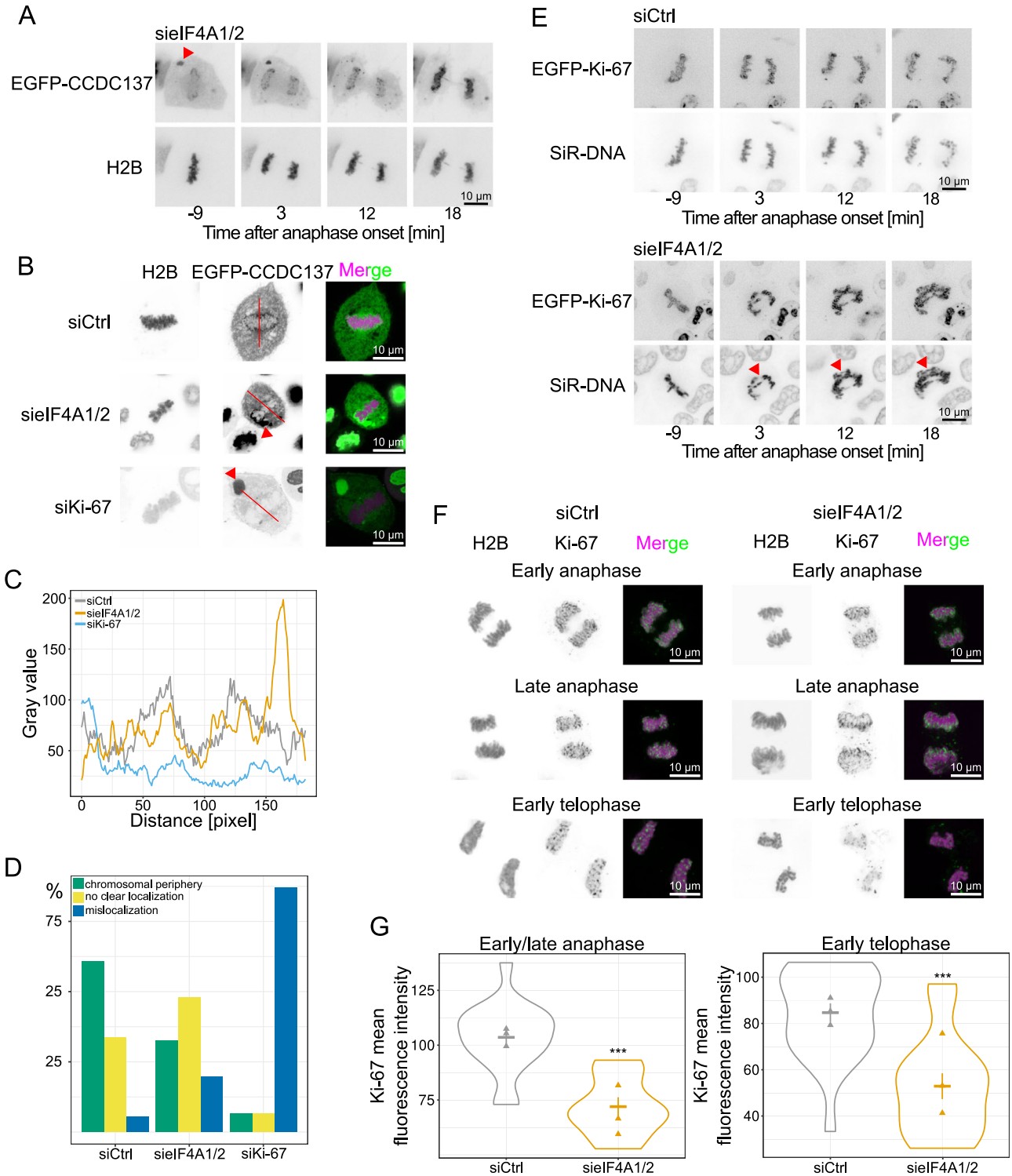

is essential for nuclear reformation[20,21]. This process necessitates the co-condensation of Ki-67 and perichromatin RNAs, leading to the formation of liquid-like condensates on the chromatin surface as mitosis concludes. It is thus plausible that the functions of eIF4A1/2 are interconnected with this mechanism by maintaining RNA on perichromatin during mitosis, thereby playing a crucial role in the transition at mitotic exit. For the proposed action as an RNA chaperone and mobilizer the RNA affinity of eIF4A1/2 should be in the same range as perichromosomal RNA binding proteins, which are currently often unknown. For DDX21 this is in the lower micromolar range (up to

30 μM, depending on the nature of RNA)[53,54] and comparable to the affinity of eIF4A1[55].

We could not detect large RNA foci following the downregulation of eIF4A1/2, unlike what has been observed in Ki-67 knockout cells[21]. This suggests that the impact of eIF4A1/2 downregulation on perichromatin integrity may be less pronounced than that caused by Ki-67 depletion, which is recognized as a central organizer of this structure[13,35]. It is possible that cytosolic droplets either escaped detection or did not form, as the critical cytosolic RNA concentration for phase separation has not been reached. However, downregulation of eIF4A1/2 expression not only reduced the RNA signal on

**Fig. 4 | Downregulation of eIF4A1/2 affects the perichromosomal layer. A** Time-lapse images of HeLa cells stably expressing H2B-mPlum and EGFP-CCDC137, transfected with a combination of 20 nM eIF4A1 and eIF4A2 siRNA oligos for 72 h showing cytosolic EGFP-CCDC137 aggregates (red arrowhead) seen in 3 independent experiments. Time is normalized to the first anaphase frame. **B** High-resolution imaging of fixed HeLa cells stably expressing H2B-mPlum (magenta in overlay) and EGFP-CCDC137 (green in overlay), transfected with 20 nM control, a combination of eIF4A1 and eIF4A2 or Ki-67 siRNA oligos for 72 h showing cytosolic EGFP-CCDC137 aggregates (red arrowheads). Line scans presented in (**C**) are indicated. **C** Line scans of the EGFP-CCDC137 fluorescence intensity in the cells shown in (**B**). Line scans are drawn across 200 pixels in each cell, and the EGFP-CCDC137 signal is presented in gray values. **D** Quantitation of EGFP-CCDC137 localization from one experiment as in (**B**) showing cell percentage with more than 90 cells per condition. **E** Time-lapse images of HeLa cells expressing EGFP-Ki-67 transfected with 20 nM control or a combination of eIF4A1 and eIF4A2 siRNA oligos for 72 h and treated with 250 nM SiR-DNA for 2 h to visualize chromatin. Time is normalized to the first anaphase frame. Red arrowheads indicate unsegregated chromatin masses seen in 5 independent experiments. **F** Ki-67 immunostaining (green in overlay) of HeLa cells stably expressing H2B-mCherry (magenta in overlay) transfected with 20 nM control or a combination of eIF4A1 and eIF4A2 siRNA oligos for 72 h at different stages of mitotic exit. **G** Quantitation of the Ki-67 mean fluorescence intensity in anaphase and telophase from three independent experiments as in **F**. The violin plots show the means of three independent experiments (triangles) each including 18 (siCtrl) and 10 (sieIF4A1/2) cells for early/late anaphase and 22 (siCtrl) and 15 (sieIF4A1/2) cells for late telophase and the overall mean ± s.e.m. Significance was tested by a two-tailed unpaired $t$ test with Welch's correction (early/late anaphase, ***$P = 1.35 \times 10^{-5}$; early telophase, ***$P = 7.95 \times 10^{-5}$). Scale bars: 10 µm. Source data are provided as a Source Data file.

perichromatin but also that of DDX18 and DDX27 signal on perichromatin but was accompanied by cytosolic foci of these proteins (Fig. 3). Likewise, eIF4A1/2 downregulation mislocalized EGFP-CCDC137 from the perichromatin toward the cytosol (Fig. 4).

eIF4A1/2 acts as a helicase during translation initiation. Interestingly, for its function in vitro decondensation, eIF4A1 does not require its ATPase activity: The E183Q mutant, which can still bind ATP and RNA but does not hydrolyze ATP[28,56,57], can replace the wild-type protein in this assay (Fig. 1D, E). In the chromatin tethering assay, the E183Q can slightly stimulate chromatin decondensation. This is likely because here eIF4A1/2 does not function as a processive helicase but rather as an RNA chaperon[52,58]. RNA duplex unwinding is coupled not so much to ATP hydrolysis as to ATP-dependent RNA binding of eIF4A1. eIF4A1 kinks the RNA locally and destabilizes duplexes, which dissociate thermally[59–62]. Accordingly, the eIF4A1 E183Q can resolve RNA-DDX18 droplets. In contrast, the R362/365Q mutant, defective in RNA binding, was inactive, indicating that for droplet resolution, RNA binding but not ATPase/helicase activity of eIF4A1 is crucial. Like for stress granules, for droplet resolution RNA binding but not ATP hydrolysis is required for eIF4A1 action[52]. Accordingly, the co-factors eIF4G and eIF4B, which significantly stimulate the only weak helicase activity of eIF4A1/2 100-fold[63–65], are not required for chromatin decondensation. This might also explain why eIF4A3, whose cellular function in splicing does not depend on a helicase activity[22], can enhance chromatin decondensation in the tethering assay and rescue upon overexpression, in part, the effect of eIF4A1/2 downregulation. As eIF4A3 is much less abundant as eIF4A1/2 (approximately tenfold lower copy numbers in U2OS cells[29]) we think that it does not play a major role in chromatin decondensation under physiological conditions. For the co-factor eIF4H, an extension of telophase is observed (Supplementary Fig. 1F). However, eIF4H downregulation does not strongly impact chromatin area increase during mitotic exit (Supplementary Fig. 1G). It is thus likely that the effect on telophase length is independent from chromatin decondensation. In line with this, chromatin tethering of eIF4H does not enhance chromatin decondensation (Supplementary Fig. 7E).

eIF4A1/2 does not enrich on mitotic chromatin, neither in immunofluorescence nor as EGFP-fusions (Fig. 5) as one might expect for a bona fide chromatin decondensation factor, and the same is true for the other factors of the eIF4F complex. However, for the proposed function of eIF4A1/2 as a cytosolic RNA chaperon, chromosome enrichment is not required, especially given the high cytosolic concentration of these helicases. However, local enrichment on chromosomes slightly enhances chromatin decondensation (Fig. 6), and this also applies to eIF4A3. It is accordingly possible that eIF4-helicases in the vicinity of the perichromatin act, keeping this liquid condensate dynamic supporting chromatin decondensation. However, in this assay, the ATPase-deficient E183Q does not show the same enhancement of chromatin decondensation as the wild-type protein upon

chromatin enrichment. It can thus not be excluded that close to the perichromatin layer, eIF4A1/2 can perform an additional ATPase-dependent function, that is different from its cytosolic chaperone activity. Of note, the tethering assays to enhance the concentration of eIF4A1/2 on chromatin rely on the addition of rapamycin (Fig. 6). As rapamycin via mTOR-inhibition also affects eIF4B and 4E-BP1 phosphorylation status[66] we can formally not exclude that the observed effect occurs via an indirect inhibition of eIF4A activity. However, we regard this possibility as unlikely given that the tethering effect observed is independent of eIF4B and that rapamycin does not impact chromatin decondensation in experiments where other eIF4F-components were tethered to chromatin (Supplementary Fig. 7).

Notably, eIF4A1 and eIF4A2 are rapidly excluded from the nucleus upon nuclear envelope reformation. It has been speculated that this helps to establish protein translation as a purely cytosolic process[38]. Considering the eIF4A1/2 function in chromatin decondensation, the rapid exclusion from the nucleus might also be functional to restrict eIF4A1/2 mediated chromatin decondensation to anaphase and early telophase. In this context, it is interesting that an eIF4A1 construct with a C-terminal nuclear localization signal slightly enhances chromatin decondensation, especially in late telophase, even though the largest fraction of the protein does not enrich in the reforming nucleus.

eIF4A1 is the most abundant DEAD-box RNA helicase in HeLa and U2OS cells[29,32], with 5 to 50 eIF4A1 molecules per mRNA estimated[52]. eIF4A1/2 is also present in a vast excess over the other factors of the eIF4F complex[32,33]. The mitotic function we describe here for eIF4A1/2 outside of the eIF4F complex and independent of its role in translational initiation might be causal for this. It is unclear whether eIF4A1/2 has a preference for certain types of RNAs or RNA structures of the chromatin surface, given that in translation initiation, eIF4A1 may preferentially act on mRNAs with purine-rich 5′UTR sequences[67]. This and the general nature of the RNA of the perichromatin layer remain open as an interesting question for further research.

## Methods

### Animal husbandry
Female *Xenopus laevis* used in this study were accommodated und used according to the German legislation regarding animal care and husbandry, approved by the animal protection commissioners of RWTH Aachen University.

### Protein expression and purification
*Xenopus laevis* eIF4A1, eIF4A2, and eIF4A3 were cloned into a pET28a vector (see Supplementary Data 1) for all constructs used in this study). The corresponding R362Q/R365Q and E183Q mutations in eIF4A1 were generated by site-directed mutagenesis. Human DDX18, DDX21, and DDX27, and *Xenopus laevis* eIF4A1, eIF4A2, and eIF4A3 were cloned into a modified pET28a vector with a N-terminal EGFP-tag or mCherry-tag. All proteins were expressed in BL21de3 *E. coli* by autoinduction in

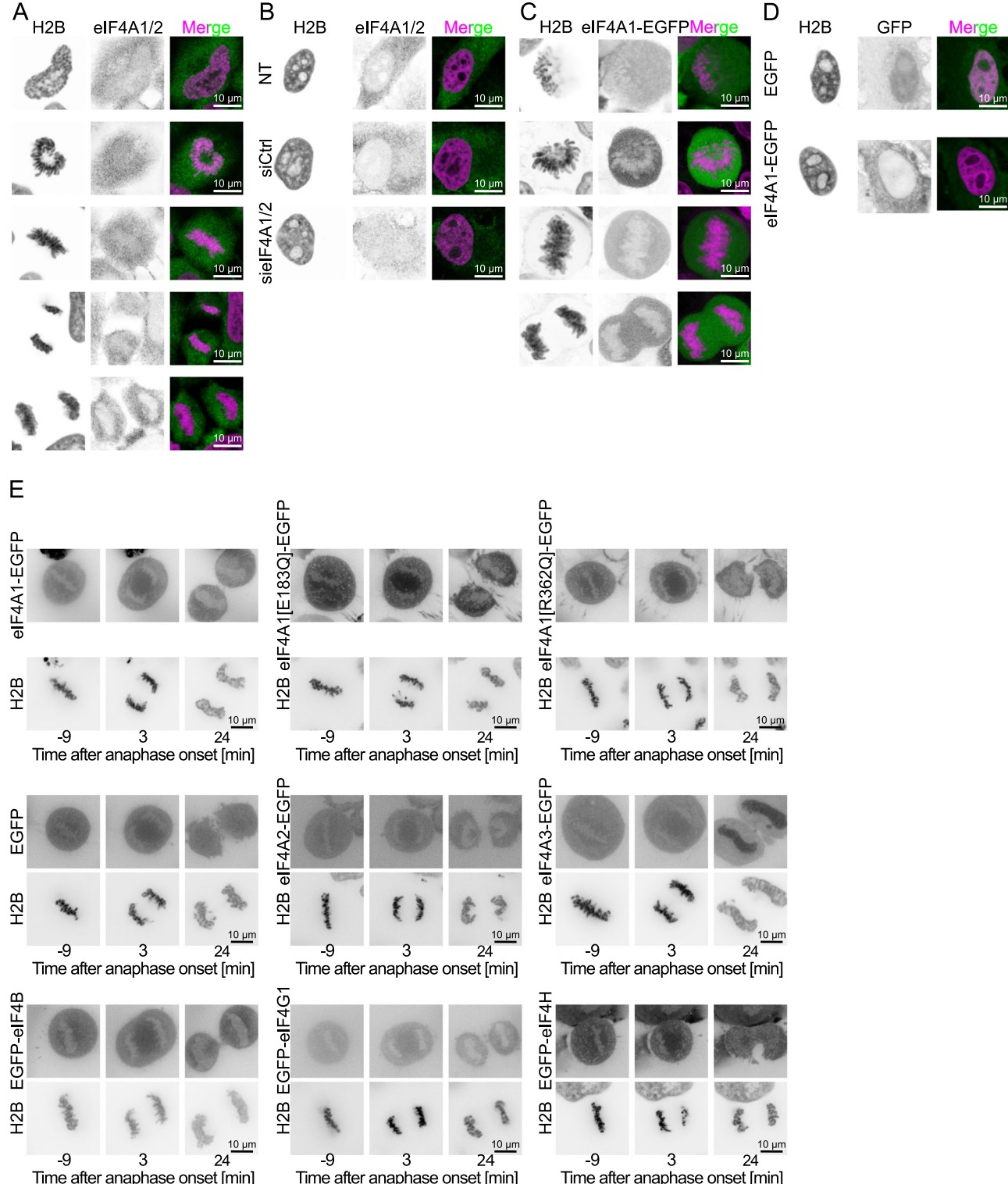

**Fig. 5 | eIF4A1/2 is rapidly excluded from the reforming nucleus. A** eIF4A1/2 immunostaining (green in overlay) of HeLa cells stably expressing H2B-mCherry (magenta in overlay) at mitotic different stages, representative for 4 independent experiments. Scale bars: 10 μm. **B** eIF4A1/2 immunostaining (green in overlay) of interphase HeLa cells stably expressing H2B-mCherry (magenta in overlay), untransfected or transfected with 20 nM control or a combination of eIF4A1 and eIF4A2 siRNA oligos for 72 h, representative for 4 independent experiments. Scale bars: 10 μm. **C** Fixed samples of different mitotic stages of HeLa cells expressing H2B-mCherry (magenta in overlay) and GFP or eIF4A1-EGFP (green in overlay),

representative for 2 independent experiments. Scale bars: 10 μm. **D** Fixed samples of interphase HeLa cells expressing H2B-mCherry (magenta in overlay) and GFP or eIF4A1-EGFP (green in overlay), representative for 2 independent experiments. Scale bars: 10 μm. **E** Time-lapse images of H2B-mCherry HeLa cells expressing eIF4A1-EGFP, the corresponding ATPase deficient mutant eIF4A1 E183Q or the RNA-binding deficient R362Q mutant, EGFP, eIF4A2-EGFP, eIF4A3-EGFP, EGFP-eIF4B, EGFP-eIF4G1 or EGFP-eIF4H, representative for 3 independent experiments per construct. Time is normalized to the first anaphase frame. Scale bars: 10 μm.

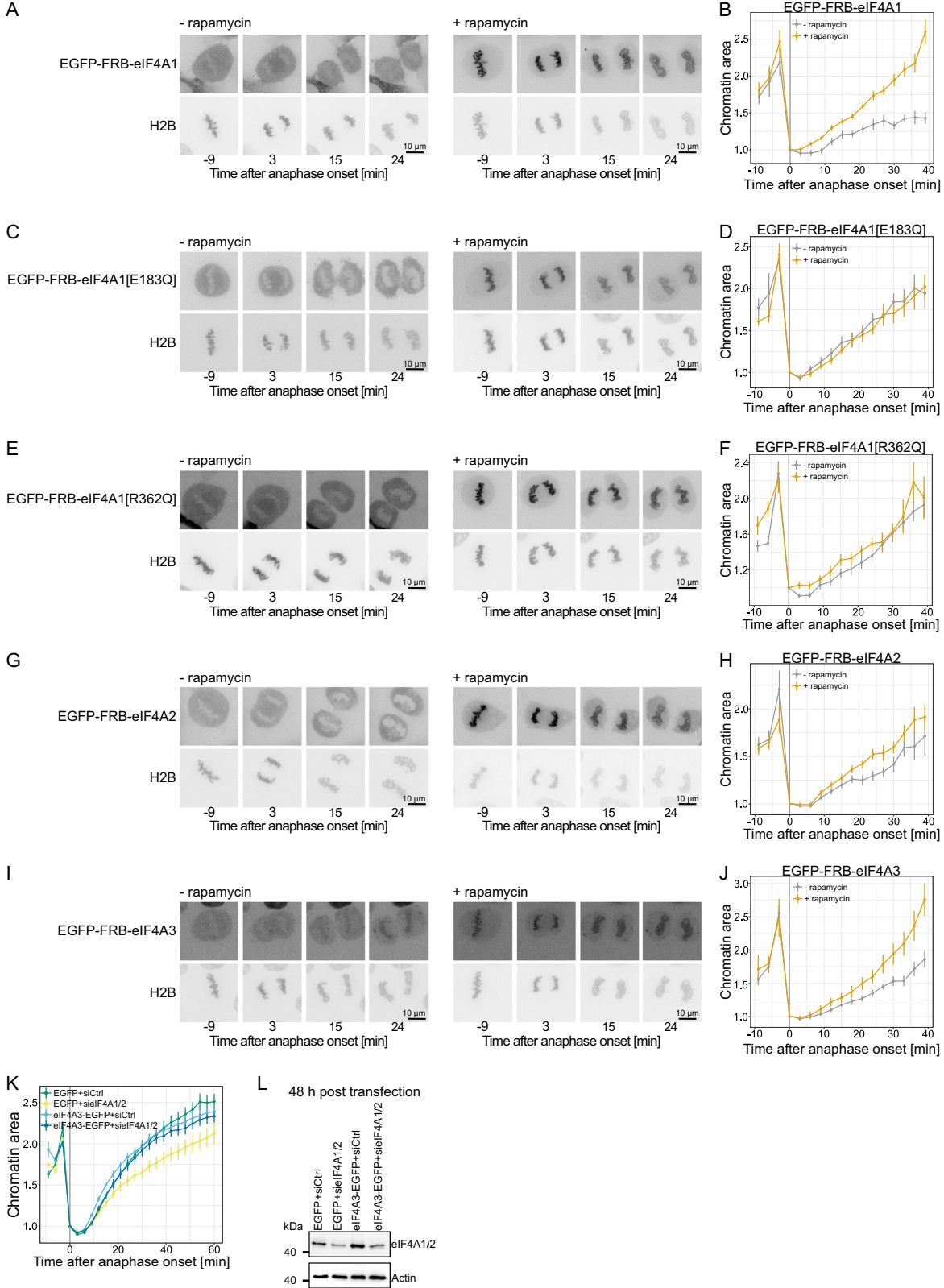

LB medium at 18 °C and purified using Ni-NTA beads (Qiagen) followed by size exclusion chromatography on a Sephadex 200 increase (10/300) column (GE Healthcare).

### Antibodies

Polyclonal *Xenopus* eIF4A1 antibodies were generated in rabbits using full-length *Xenopus laevis* eIF4A1, expressed from a pET28a vector, see above, and employed at a dilution of 1:1,000 for Western blotting. Antibodies against human eIF4A1 (Cell Signaling, #2490, 1:1000), eIF4A1/A2 (Santa Cruz, sc-377315, 1:500), eIF4A3 (abcam, ab32485, 1:1000), eIF4B (Cell Signaling, #13088, 1:1000), eIF4E (Cell Signaling, #2067, 1:1000), eIF4G1 (Cell Signaling, #2498, 1:1,000), eIF4H (Cell Signaling, #3469, 1:1000) Mab414 antibody for detection of Nup62 (Biolegend, 902902, 1:5000) and actin (MP Biomedicals, 691001,

**Fig. 6 | Tethering of eIF4A1/2 to chromosomes accelerates chromatin decondensation. A** Time-lapse images of H2B-mPlum-FKBP HeLa cells expressing EGFP-FRB-eIF4A1 after adding 200 nM DMSO (control, - rapamycin) or rapamycin. Time is normalized to the first anaphase frame. Scale bars: 10 μm. **B** Time-dependent quantitation of the chromatin area of the experiments as in (**A**), normalized to the first anaphase frame. Dots represent mean ± s.e.m. from 12 (-rapamycin) and 14 (+ rapamycin) cells. **C** Same as in (**A**) but with the ATPase deficient mutant of eIF4A1 E183Q. **D** Same as in (**B**) but with the ATPase deficient mutant of eIF4A1 E183Q from 7 (-rapamycin) and 9 (+ rapamycin) cells. **E** Same as in (**A**) but with the RNA-binding mutant of eIF4A1 R362Q. **F** Same as in (**B**) but with the RNA-binding mutant of eIF4A1 R362Q from 9 (-rapamycin) and 10 (+ rapamycin) cells. **G** Same as in (**A**) but with EGFP-FRB-eIF4A2. **H** Same as in (**B**) but with EGFP-FRB-eIF4A2 with 15

(-rapamycin) and 17 (+ rapamycin) cells. **I** Same as in (**A**) but with EGFP-FRB-eIF4A3. **J** Same as in (**B**) but with EGFP-FRB-eIF4A3 with 10 (+ rapamycin) and 10 (-rapamycin) cells. **K** Time-dependent quantification of the chromatin area normalized to the first anaphase frame. HeLa cells stably expressing H2B-mCherry were co-transfected with GFP or eIF4A3-GFP and 40 nM control or a combination of eIF4A1 and eIF4A2 siRNA oligos for 48 h. Dots represent mean ± s.e.m. from each condition with 53 (EGFP + siCtrl), 37 (EGFP + sieIF4A1/2), 71 (eIF4A3-EGFP + siCtrl) or 45 (eIF4A3-EGFP + sieIF4A1/2) daughter chromatin masses. **L** Western blot showing the downregulation of eIF4A1/2 at 48 h post-transfection with 40 nM siRNA oligos and EGFP or eIF4A3-EGFP in HeLa cells stably expressing H2B-mCherry. Samples were analyzed with antibodies recognizing human eIF4A1/2. Actin serves as loading control. Source data are provided as a Source Data file.

1:10,000)) were employed for Western blotting. Antibodies against human eIF4A1/A2 (Santa Cruz, sc-377315, 1:100), Ki-67 (Millipore, Mab4190, 1:100), DDX18 (bethyl laboratories, A300-636A, 1:200, methanol fixation) and DDX27 (Santa Cruz, sc-81074, 1:200, methanol fixation) were employed for immunofluorescence.

For Western blotting, goat anti-rabbit IgG-HRP and goat anti-mouse IgG-HRP (Calbiochem) were used at a dilution of 1:5000. Signals were detected using Western Bright Quantum (Advansta) as an ECL substrate on an ImageQuant LAS-4000 system (Fuji). Intensities of the non-saturated signals were analyzed using (Fiji Is Just) ImageJ (version 1.54 g)[68]. The bands to be measured were defined as the region of interest, and the mean intensity signals of these regions were extracted. Background intensity was subtracted to calculate the band signal.

Donkey anti-rabbit IgG Alexa Fluor 405, goat anti-mouse IgG Alexa Fluor 488, goat anti-rabbit IgG Alexa Fluor 546, goat anti-mouse IgG Alexa Fluor 647, goat anti-rabbit IgG Alexa Fluor 647 (Invitrogen) were used at a dilution of 1:2000 for immunofluorescence. The Ki-67 signal was quantified using a custom Python (version 3.11.1) script (http://www.python.org).

### In vitro assays

High-speed *Xenopus laevis* egg interphase extracts were prepared from A23187 ionophore-activated eggs and mitotic chromatin clusters from metaphase-arrested HeLa cells[19,46]. For in vitro chromatin decondensation approximately 1000 mitotic chromatin clusters were incubated in 20 μl of high-speed egg extracts, supplemented with 3 μM 6-dimethylaminopurine, 10 mM ATP, 10 mM creatine phosphate, 0.2 mg/ml creatine kinase, and 0.4 mg/ml glycogen at 20 °C for 90 min[19,46]. For immunodepletion, Protein-A Sepharose FF (GE Healthcare) with covalently linked eIF4A1 or control antibodies were generated using Dimethylpimelimidat[69]. For add-back experiments, recombinant *Xenopus* eIF4A1 wild type or mutant proteins, eIF4A2 or eIF4A3 (all dialyzed to 250 mM sucrose, 50 mM KCl, 10 mM HEPES-KOH, 2.5 mM MgCl$_2$) were added to eIF4A1/2-depleted extracts to approximately endogenous levels, based on the determined protein concentration (BCA method) and confirmed by Western blotting. For inhibition experiments, RNA was added to a final concentration of 0.05 mg/ml, and RNase A (NEB) to a final concentration of 50 μg/ml. After incubation for 90 min at 20 °C, samples were fixed in 4% paraformaldehyde and 0.5% glutaraldehyde in 80 mM PIPES pH 6.8, 1 mM MgCl$_2$, 150 mM sucrose, and 10 μg/ml DAPI and the chromatin reisolated by centrifugation through a 30% (w/v) sucrose cushion on poly-L-lysine-coated coverslips and mounted in Vectashield (Vector Laboratories)[19,46]. For analysis, fluorescence images were acquired using a Zeiss LSM710 confocal microscope equipped with a Plan-Apochromat 63 × / 1.4 immersion oil objective and 405, 488, and 561 nm lasers using ZEN software. The border smoothness of the chromatin masses was used to quantify chromatin decondensation[19,46]. For this, best in-focus slices from at least 40 random nuclei from three independent experiments per condition were converted to 8-bit TIFF files in Fiji, and the chromatin morphology was analyzed with a custom

Python script using OpenCV. Background noise was removed with a median filter (cv2.medianBlur), Otsu thresholding was performed to initially segment the chromatin masses (cv2.threshold), and binary holes in the masses were filled with morphological operations (ndimage.morphology.binary_fill_holes). A version of the segmented chromatin with softer borders was calculated by applying a median filter. Border smoothness was calculated as a ratio between the intersection and the union of the segmentation and the soft-border version areas.

For visualizing RNA in the in vitro condensation assay, mitotic chromatin clusters from HeLa cells stably expressing EGFP-H2B[70] were generated[19,46], and assays were performed in the presence of 5 μg/ml Alexa-546 labeled RNA and where indicated 50 μg/ml unlabeled RNA. After 90 min incubation, the samples were directly analyzed without fixation on a Zeiss LSM710 confocal microscope equipped with a Plan-Apochromat 63 × / 1.4 immersion oil objective and 488 and 561 nm lasers, using ZEN software.

For in vitro phase separation, 50 mg of yeast total RNA (Roche, 10109223001) was dissolved in 1 ml 10 mM HEPES, pH 7.5, 1 mM DTT and heated for 2 min at 95 °C. After cooling on ice and clearance by centrifugation (10 min at 15,000 g) 100 μl of supernatant was supplemented with 25 μl 10 mg/ml BSA in 10 mM HEPES, pH 7.5, 1 mM DTT, 20 μl RNasin (40 U/μl, Promega; N251A) and 105 μl 10 mM HEPES, pH 7.5, 1 mM DTT to yield a 4 mg/ml RNA stock solution. For labeling, 100 μl of 1 mg/ml total yeast RNA, heated and cooled as above, was supplemented with 100 μl 400 mM NaHCO$_3$ pH 8.3, followed by 10 μl of 13 mg/ml Alexa-546-Succinimidylester in DMSO. The solution was incubated at 25 °C for 2 h, slightly shaking. The reaction was quenched by adding 20 μl of 1.5 M hydroxylamine (pH 8.5) for 1 h at 25 °C and the labeled RNA was separated from the free dye by passage over a Sephadex G-25 column (NAP-10 column, Cytiva). Labeling efficiency was determined by taking a UV spectrum of the unlabeled and labeled RNA.

In vitro phase separation assays were performed in 2 μl reactions containing 1 mg/ml RNA, 10 μM recombinant EGFP-DDX18, EGFP-DDX21, EGFP-DDX27, EGFP-eIF4A1, EGFP-eIF4A2, or EGFP-eIF4A3 and, as indicated 0, 1, 3, 10 or 30 μM of recombinant untagged wild type or mutant eIF4A1, eIF4A2, or eIF4A3 in a buffer containing 10 mM HEPES, pH 7.5, 1 mM DTT, 1 mM ATP, 150 mM NaCl, 0.25 μg/μl BSA, 0.1 U RNasin were put on 8 well diagnostic microscope slide (Thermo Scientific, ER-201B-CE24), covered glass slide and immediately analyzed on a Zeiss LSM710 confocal microscope equipped with a Plan-Apochromat 63 × / 1.4 immersion oil objective and 488 and 561 nm lasers, using ZEN software. For testing co-enrichment in droplets, 2 μl reactions contained 1 mg/ml RNA, 10 μM recombinant EGFP-DDX18, EGFP-DDX21, or EGFP-DDX27 and 0.5 μM mCherry-eIF4A1 or mCherry-DDX27. For visualizing RNA, the unlabeled and Alexa-546 labeled RNA was mixed in a 9:1 molar ratio immediately before setting up the phase separation experiment.

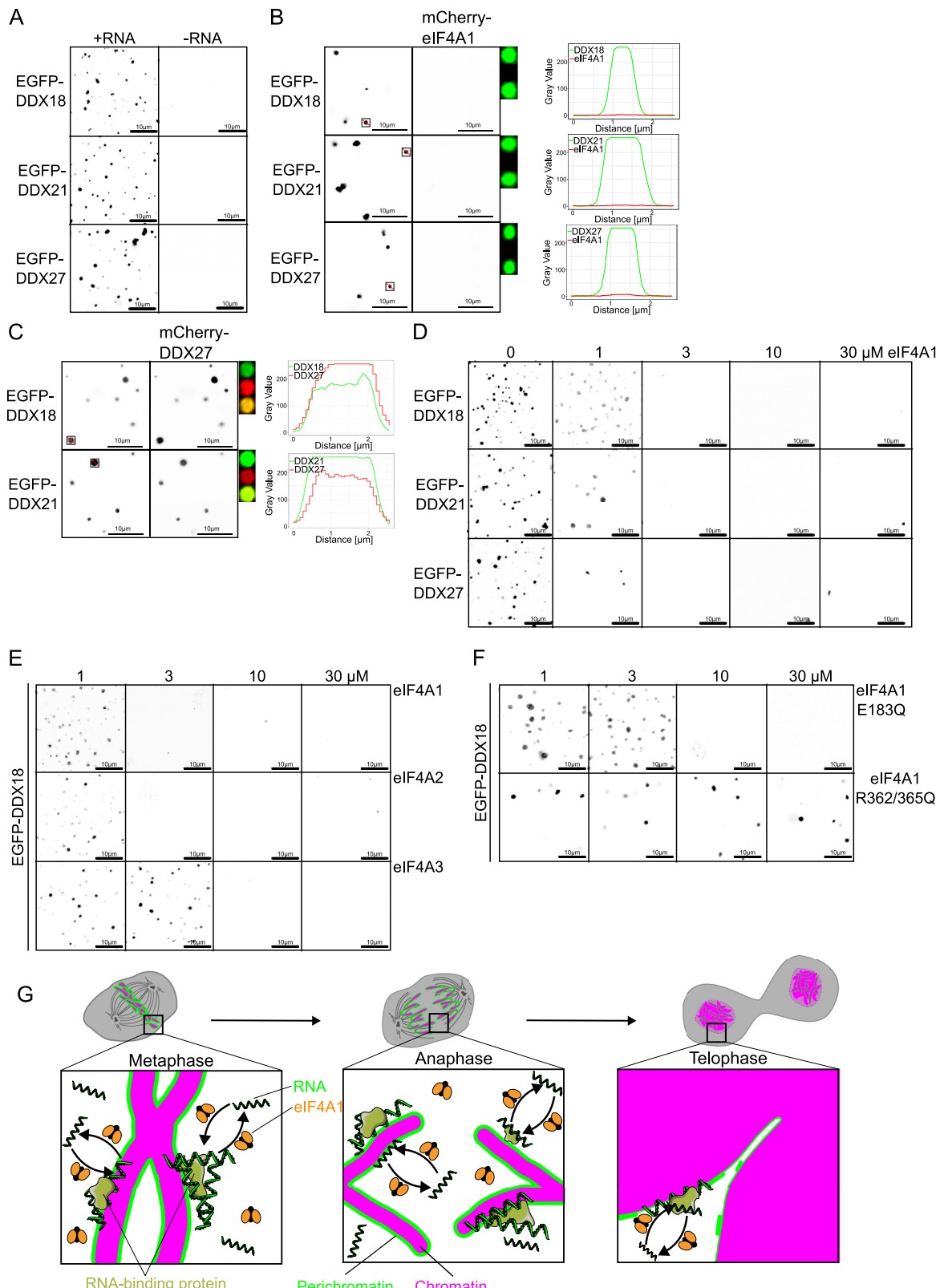

## Cellular experiments

For visualizing in cells, human eIF4A1, eIF4A2, eIF4A3, eIF4B, eIF4G1, eIF4H, and CCDC137 were cloned into an EGFP-C3 vector and/or EGFP-N1 generating N- or C-terminally EGFP-tagged protein fusions (Invitrogen) (see Supplementary Data 1 for complete list). The corresponding E183Q and R362Q mutants of eIF4A1 were generated in the EGFP-N1 vector. The SV40 large antigen-based nuclear localization signal (NLS, aa sequence PKKKRKV) was integrated into wild-type and mutant EGFP-C3-eIF4A1 C-terminal of the EGFP. For chromatin tethering, eIF4A1, eIF4A2, eIF4A3, eIF4B, eIF4G1, and eIF4H were cloned into a modified EGFP-C3 vector downstream of the EGFP and FRB domains. For siRNA resistant version of EGFP-N1-eIF4A1 (pEGFP-N1-eIF4A1siRES), the oligo targeting sequence

**Fig. 7 | eIF4A1/2 destabilizes RNA condensates. A** In vitro phase separation of 10 µM recombinant EGFP-DDX18, EGFP-DDX21, or EGFP-DDX27 in the presence or absence of 1 mg/ml RNA. Confocal images are representative of three independent experiments. **B** In vitro phase separation of 10 µM recombinant EGFP-DDX18, EGFP-DDX21, or EGFP-DDX27 (green in the presence of 1 mg/ml RNA and 0.5 µM mCherry-eIF4A1. Confocal images are representative of three independent experiments. Single RNA condensates are marked and shown as a close-up in the EGFP (green), mCherry (red), and merged channel (yellow as an overlay of red and green). On the right, line scans through a condensate is shown (EGFP signal in green, mCherry signal in red). Scale bars: 10 µm. **C** In vitro phase separation of 10 µM recombinant EGFP-DDX18 or EGFP-DDX21 and mCherry-DDX27 in the presence of 1 mg/ml RNA. Confocal images are representative of three independent experiments. Single RNA condensates are marked and shown as a close-up in the EGFP (green), mCherry (red), and merged channel (yellow as an overlay of red and green). On the right line, scans through a condensate is shown (EGFP signal in green, mCherry signal in red).

**D** In vitro phase separation in the presence of RNA of the experiments as in (**A**) with increasing concentration of recombinant eIF4A1. Confocal images are representative of three independent experiments. **E** In vitro phase separation of 10 µM recombinant EGFP-DDX18 in the presence of 1 mg/ml RNA with increasing concentration of recombinant eIF4A1, eIF4A2, or eIF4A3. Confocal images are representative of three independent experiments. **F** In vitro phase separation of 10 µM recombinant EGFP-DDX18 in the presence of 1 mg/ml RNA with increasing concentration of the recombinant ATPase deficient E183Q or RNA-binding deficient R362/365Q mutant of eIF4A1. Confocal images are representative of three independent experiments. **G** Working model of eIF4A1/2 function as RNA mobilizers: As cytosolic RNA chaperone, eIF4A1/2 facilitates the formation and/or maintenance of the perichromatin layer by enhancing RNA dynamics, thereby promoting their integration into the perichromatin layer. Scale bars: 10 µm. Source data are provided as a Source Data file.

---

(ACAUCAACGUGGAACGAGA) was replaced by a sequence with a different codon usage (ATATTAATGTCGAGCGGGA).

HeLa cells stably expressing H2B-mCherry[71], H2B-mPlum-FKBP[39], EGFP-Ki-67[14], H2B–mCherry and IBB–EGFP[34], as well as RPE1 cells, stably expressing H2B-RFP were grown in Dulbecco's modified Eagle's medium (DMEM; Gibco Life Technologies, 31966021) with pyruvate supplemented with 10% fetal bovine serum (FBS; Gibco, 10500064) and 1% penicillin-streptomycin (Gibco, 15140122) at 37 °C in a 5% $CO_2$ atmosphere. For generating a stable CCDC137-expressing cell line, HeLa cells stably expressing H2B-mPlum-FKBP were transfected with human EGFP-CCDC137, selected with geneticin at a final concentration of 500 µg/ml (Gibco, 10131035) and FACS sorted. All cell lines were regularly verified negative for mycoplasma contamination and are listed in Supplementary Data 1.

siRNA oligos were obtained from Qiagen or Invitrogen (see Supplementary Data 1 for a complete list). RNAi oligos were reverse transfected with the lipofectamine RNAi MAX transfection reagent (Invitrogen, 13778075) and used at 20 nM for 24 h to 96 h. For the double knockdowns of eIF4A1 and eIF4A2 end concentrations were 10 nM per RNA oligo. Lipofectamine and siRNA were mixed in Opti-MEM (Gibco), incubated for at least 20 min at room temperature, and put in a well. 260 µl of cell suspension in growing medium was added to 40 µl of siRNA solution to obtain a final concentration of 20 nM of siRNA and 8000 cells per well. Cells were incubated for 72 h at 37 °C in a 5% $CO_2$ atmosphere.

Plasmid transfections were performed using jetPRIME transfection reagent (Polyplus-Sartorius, Göttingen, Germany). Cells were seeded at a density of 8000 cells per well 24 h before plasmid transfection. For 150 ng plasmid DNA, 0.3 µl transfection reagent was used, and the medium was changed after 4 h. Cells were incubated for 24 h at 37 °C in a 5% $CO_2$ atmosphere. For co-transfections of plasmid DNA and siRNA oligos (Figs. 2G, H; 6K, L), both reagents were diluted in transfection buffer, mixed with JetPrime transfection reagent, incubated for 10 min, and added dropwise on cells seeded the day before. 8000 cells were treated with 75 ng DNA and 40 nM siRNA for 48 h at 37 °C in a 5% $CO_2$ atmosphere.

To analyze the RNA signal after eIF4A1/2 knockdown, HeLa H2B-mCherry cells were transfected as described above. 24 h to 72 h post-transfection cells were synchronized by a double thymidine block. In detail, cells were incubated for 24 h in a medium containing 2 mM thymidine (Millipore, 6060-5GM), followed by 3x wash with warm PBS for release. After 8 h, the medium was exchanged again for 2 mM thymidine-containing medium and incubated for 16 h, followed by 3x wash with warm PBS and release for 5 h. For labeling of nascent RNA, cells were then incubated with 200 µM 5-ethynyl uridine for 3 h (Click-iT RNA imaging Kits, Invitrogen, C10329), followed by 3x wash with PBS and fixation with 4% PFA. After 2x wash with PBS, cells were permeabilized with PBS containing 0.5% Triton X100. For RNA staining,

cells were treated with Alexa Fluor azide 488 in a reaction cocktail for 1 h. Afterward, cells were washed with rinse buffer and mounted with Mowiol.

For tethering experiments and labeling of perichromatin proteins, HeLa H2B-mPlum-FKBP cells were transfected with EGFP-FRB-eIF4I. The next day, cells were treated with 200 nM rapamycin for 20 min, followed by fixation and immunostaining. For quantitation of RNA or DDX18/Ki-67 on chromatin, Fiji was used. Default thresholding was applied to the tiff images in the chromatin channel, and the resulting mask was used to measure the mean pixel intensity in the second channel. For background subtraction, a circular ROI was randomly placed in the cytoplasm of each cell to measure mean pixel intensity.

### Live cell imaging

For live cell imaging, cells were grown in IBItreat chambers with polymer coverslips (IBIDI, Gräfelfing, Germany, 80826) at 37 °C and 5% $CO_2$. High-resolution imaging was performed on a spinning disk confocal microscope Ti2 Eclipse (Nikon) equipped with an LED light engine SpectraX (Lumencor, Beaverton, Oregon, USA) and GFP/mCherry/Cy5 filter sets, a Lambda Oil 60 × NA 1.4 objective, and environmental control system UNO-T-H-CO2 (Okolab, Ottaviano, NA, Italy). AR-Elements software (Nikon) equipped with a software-based autofocus module was used to perform confocal fluorescence imaging of the single best-in-focus optical section in metaphase every 2 or 3 min for approximately 1 h. The same microscope was used in a wide field mode for long-term lower-resolution imaging with the 10 × 0.5 NA, the 20 × 0.75 NA, or the 40 × 0.95 NA Lambda Air objective and a 3 min interval.

HeLa cells stably expressing H2B-mCherry and IBB-EGFP[34] were transfected with the indicated siRNAs. After 48 h, the cells were subjected to live cell imaging. Using Fiji, random mitotic cell trajectories with clear nuclear IBB-EGFP labeling before mitotic entry were identified visually according to the chromatin morphology. Ten trajectories were collected per condition of two independent experiments. Then, the records were visually inspected to determine the time of recruitment of IBB-EGFP to the nuclear compartment after the last metaphase frame (Supplementary Fig. 3B).

The quantitation of the dwell time in the different mitotic phases on the long-term lower-resolution imaging experiments (10x and 20x) for Fig. 2A, Supplementary Fig. 1A–D, F, and Supplementary Fig. 3D was performed using CecogAnalyzer 1.5.2 as in refs. 30,71. The machine learning algorithm was trained on the global differential morphologies of the chromatin throughout the cell cycle. Here, interphase, prophase, metaphase, and anaphase were scored according to the classical chromatin cytology of mitosis. Chromatin masses that followed anaphase events and were distinct from the anaphase and interphase chromatin morphologies of cells in the same field of view were annotated as telophase. A custom script in R (version 4.4.0)[72] obtained

mitosis dwell times by unbiasedly discarding erroneous trajectories. Briefly, the script filters the CellCognition numerical output tables for trajectories whose parameters are unambiguously mitotic. The frequency and type of chromatin segregation errors were then evaluated by visual scoring and classification on image galleries of valid mitotic trajectories.

The analysis of the chromatin features, such as the chromatin area during mitotic exit and interphase nuclear area, on the long-term lower-resolution imaging experiments for Fig. 2B and Supplementary Fig. 1E, G was performed with LiveCellMiner v1.00 31. The software automatically extracted and synchronized in anaphase onset mitotic trajectories with a length of 90 min before and 180 min after anaphase onset. The erroneous tracks that do not contain a mitotic event were discarded by a trainable LSTM network that assesses each trajectory's validity based on a few annotations.

For the analysis of the chromatin area during mitotic exit on long-term lower-resolution live cell imaging experiments (recorded with a 40x objective, Figs. 2G and 6K), random individual cell mitotic trajectories starting in the last metaphase were identified visually, manually extracted from large multidimensional microscopy files and converted in stacked ome.tiff files containing single-cell mitotic trajectories using the AR-Elements software (Nikon). A custom Python script (Supplementary Data 2) was used to segment the chromatin by applying Otsu thresholding. Segmentations were measured, and the values were exported as csv-tables. Mitotic time points were calculated with a custom R script (Supplementary materials). The first ana-/telophase frame, where the first-time segmentation of two daughter cells was possible, was set as $t = 0$ min.

The analysis of the chromatin area and mean fluorescence (mean pixel density) during mitotic exit on short-term high-resolution live cell imaging (recorded with a 60x objective) for Figs. 2F, 6B, D, F, H, J Supplementary Figs. 2, 4D and E, 5, 6C, and 7 was done with a custom script written in Fiji. Briefly, chromatin was segmented by applying a Gaussian Blur ($\sigma = 1.5$) thresholder and converted to a binary mask. Mitotic phases were classified according to the sizes of the chromatin masses. Measurements of the region of interest were exported as csv-tables, and mitotic time points were calculated using a custom R script. The first ana-/telophase frame, where the first-time segmentation of two daughter cells was possible, was set as $t = 0$ min. The interphase nuclear area in Supplementary Fig. 6D was measured accordingly, analyzing only interphasic cells in the mitotic trajectories.

For translation inhibition, cycloheximide or harringtonine was added at a final concentration of 50 µg/ml or 2 µg/ml, respectively, to metaphase cells, and cells were imaged during mitotic exit. To stain DNA, live SiR-DNA (Spirochrome, Stein am Rhein, Switzerland) was used at a final concentration of 250 nM, and cells were incubated for 1 h at 37 °C in a 5% $CO_2$ atmosphere before imaging. For chromatin-tethering experiments, rapamycin was added at a final concentration of 200 nM to metaphase cells, and cells were imaged during mitotic exit.

### Statistical analyses and data reporting

No statistical methods were used to predetermine the sample size. Data were tested for normality with the Shapiro-Wilk test. The appropriate tests were chosen as follows: unpaired normally distributed data were tested with a two-tailed $t$ test with Welch´s correction ($\alpha = 0.05$). Unpaired not-normally distributed data were tested with a two-tailed Mann-Whitney U test ($\alpha = 0.05$). Scoring of cytoplasmic foci (Fig. 3F, J) was tested with a two-tailed Fisher's exact test ($\alpha = 0.05$). Time-dependent analyses of chromatin area or mean fluorescence intensity were analyzed according to refs. 16,20. All data analyses and visualization were followed whenever possible as suggested by ref. 68.

All statistical analyses were performed in R using the package *ggstatsplot* (version 0.12.4). To minimize potential human bias,

experiments were automatically analyzed by Fiji, Python, or MATLAB scripts.

### Processing of figures and videos

The contrast was linearly adjusted for visual representation in all figures and assembled using Fiji. All plots were generated in R using the package *ggplot2* (version 3.5.1), except for line plots in Fig. 2B and Supplementary Fig. 1E, G which were directly exported from LiveCellMiner. All figures were assembled and created using Inkscape 1.3.2 (Free Software Foundation, Inc. Boston, USA). Color-blind panels were adapted from ref. 69 and from the R package *viridis* (version 0.6.5). Protein schemes in the model in Fig. 6G were adapted from bioicons.com.

### Reporting summary

Further information on research design is available in the Nature Portfolio Reporting Summary linked to this article.

## Data availability

Raw microscopy data are available from the corresponding authors upon request given its large file size. Source data are provided in this paper.

## Code availability

All image analyses steps performed in this study are described in detail in the Methods section, and the code is provided in the Supplementary Data 2 file. Supplementary Data 3 provides information on the usage of the individual scripts in Python and ImageJ.

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

## Acknowledgements

This work was supported by the Confocal Microscopy Facility and the Flow Cytometry Facility, core facilities of the Interdisciplinary Center for Clinical Research (IZKF) Aachen within the Faculty of Medicine at RWTH Aachen University and by the German Research Foundation (DFG), project ID 439895892. We are thankful to Susanne Krapp for her help with transfections. This work was supported by grants from the German Research Foundation (project ID 107479298, to W.A.), by the START-Program of the Faculty of Medicine RWTH Aachen University (to R.J.), by the Habilitation Program of the Faculty of Medicine RWTH Aachen University (to R.J.) and by grants from the Fonds of the Chemical Industry (to R.J.).

## Author contributions

Conceptualization: R.J., D.M.-A., and W.A.; formal analysis: R.J., S.C.W., A.S., T.S., M.W.G.S., and D.M.-A.; investigation: R.J., S.C.W., A.S., T.S., C.S., C.L.B., R.S., M.W.G.S., and D.M.-A.; resources: I.B., S.R., R.S., M.T.-N., B.L., H.L., and J.-F.P.-C.; writing - original draft preparation: W.A.; writing - review and editing: R.J., S.C.W., D.M.-A. and W.A.; visualization: R.J., S.C.W., and D.M.-A.; supervision: R.J., D.M.-A., and W.A.; project administration: R.J. and W.A.; funding acquisition: R.J. and W.A. All authors have read and agreed to the submitted version of the manuscript.

## Funding

## Competing interests

The authors declare no competing interests.
