## [Transparent Peer Review file · Nature Communications]

The DEAD-box helicase eIF4A1/2 acts as RNA chaperone during mitotic exit enabling chromatin decondensation

Corresponding Author: Professor Wolfram Antonin

Version 0:

Reviewer comments:

Reviewer #1

(Remarks to the Author)

Jühlen, Wiesmann et al. demonstrate that eIF4A1/2, known as translation factors, facilitate chromosome decondensation in late mitosis through their RNA binding activity. They propose that eIF4A1/2 induce the dissociation of RNA from perichromatin by disrupting RNA-protein condensates.

This paper will be of great interest to researchers in the field, and I find it exciting due to its significance in advancing our understanding of chromosome decondensation during the transition from mitosis to interphase. However, several issues need to be addressed to ensure that the claims and the model are adequately supported by the data.

Major point

There is not enough data to support the proposed model

The authors argue that protein-RNA condensates are dissociated from perichromatin by eIF4A1/2 (Fig. 6G). However, some inconsistencies need to be addressed, and additional data are required to support/modify this model.

Firstly, while the authors clearly show that RNA-protein (DDX18, DDX21, or DDX27) condensates are disrupted by the addition of eIF4A1/2/3 (Fig. 6A-D), they have not checked for the dissociation of these proteins from perichromatin. It is expected that dissociation would be observed with the add-back of eIF4A1/2 and retention with depletion. Confusingly, with eIF4A1/2 knockdown, CCDC137, used as a perichromatin marker, formed clusters and moved away from the mitotic chromosome, and the signal for Ki67, the central organizer of perichromatin, also weakened (Fig. 3C and H). How can this be explained?

Secondly, the authors demonstrate that eIF4A1/2 knockdown leads to the retention of RNA on perichromatin (Fig. 3B). However, to support the proposed model, the authors need to test whether eIF4A1/2 actively dissociates RNA from perichromatin. Employing the rapamycin system shown in Fig. 5 would be useful. Additionally, when quantifying RNA at perichromatin, the authors need to measure the signal of RNA co-localized with EGFP-CCDC/Ki67. Otherwise, the results are not conclusive.

Lastly, it seems challenging to demonstrate that the findings from the condensate assay are also occurring at perichromatin. The essence of the decondensation mechanism proposed by the authors appears to lie in the RNA-binding affinity of eIF4A1/2. An additional experiment to consider would be to investigate whether eIF4A1/2 has a stronger binding affinity for RNA compared to other RNA-binding proteins localized to perichromatin. If eIF4A1/2 can competitively exclude RNA from these proteins, it may be possible to propose a model more consistent with the data and suggest a link to phase separation mechanisms by referring to previous studies. This may temper the authors' claims, but the significance of the findings remains substantial.

Minor points

1. Title suggestion: Should it be changed to eIF4A1/2 instead of eIF4A1?
2. Some figures are too small, especially Fig3A, C, D, and G. The chromosome/protein localization indicated by the arrowheads cannot be identified.
3. Please provide the author's explanation for the prolonged telophase observed with eIF4H knockdown.
4. What type of RNA is used in the condensate assay (Fig6 A-D)? Please specify in the methods section.

Reviewer #2

(Remarks to the Author)

Jühlen et al. present data in support of their hypothesis that eIF4A1 plays a role in driving de-condensation of chromatin during mitotic exit. Using *Xenopus* chromatin clusters they demonstrate that recombinant eIF4A can promote decondensation and build on this to demonstrate that RNA binding, but not helicase activity is necessary for this activity. They conclude that eIF4A is functioning as an RNA chaperone rather than a helicase to promote this activity. Overall all, the data is of high quality and largely supports their conclusions. I have a couple comments that should be addressed before publication with point #5 being the most relevant.

1. Figure 1: I realize these are in vitro assays in which recombinant proteins were added. However, it would be nice to have some loading control for these experiments. Particularly in light of the depleted lysate in which there is no band present.
2. Figure 2: Again, a minor point, but demonstrating depletion efficiency for the experiment in figures 2A – C is necessary. Is it the same as shown in 2D? If so, this should be presented 1st. If not, the data for this experiment should be shown.
3. The authors keep referring to eIF4F, but only every analyze eIF4G and eIF4A. These experiments should be conducted with eIF4E to formalize their statement that eIF4F is not necessary. Alternatively, the wording should be altered.
4. The tethering experiments are intriguing. However, I have some concerns about the use of rapamycin in this experiment as eIF4A activity is inhibited by rapamycin. Granted, this is thought to be through eIF4B phosphorylation and is likely not involved given the authors have demonstrated that the effect they see is independent of helicase activity or eIF4B. However, this potential complication should be discussed.
5. The ability of eIF4A to resolve helicase granules is interesting, however, I'm not convinced what this is showing in terms of decondensation of chromatin. Is it previously established that RNA:helicase complexes are involved in maintaining the perichromatin? I understand that helicases are implicated in LLPS resolution, but is there any connection between these helicases and perichromatin? If so, this could be explained better. As currently presented, its an interesting observation, but unclear as to how it relates. Further, this assay is showing that eIF4A3 is able to dissolve these condensates. This is in contrast to previous data demonstrating that eIF4A3 is not involved in decondensation. This further lessens the impact of this figure.

Reviewer #3

(Remarks to the Author)

The process of chromosome decondensation during telophase of mitosis is important for eukaryotes undergoing open mitosis to establish interphase chromatin structure. However, little is known about the molecules involved in this process. This paper discovered eIF4A1 as a protein involved in this process. eIF4A1 is one of the components of the eIF4F complex that works in the initiation of translation, and has ATPase activity and RNA helicase activity when working in translation initiation. However, this paper clarified using the cell-free system, in vivo (in cell) experiments, and in vitro experiments that these activities are not required for chromosome decondensation activity, and that RNA binding ability plays a key role instead. This paper is excellent in that it has discovered a previously unknown factor involved in decondensation. Moreover, it is highly novel that a factor involved in the initiation of protein synthesis has such a function completely independent of its activity. The experimental methodology is sound, and I mostly agree with the conclusions drawn from the results.

I have a few comments that should be revised before publication.

Minor comments

1. line 56, "during anaphase and early telophase": I recommend removing the word "anaphase" from this sentence. The reason is that chromosome decondensation occurs with nuclear envelope (NE) reformation, but in anaphase, NE reformation does not occur, so there is no need to include anaphase. This paper does not observe the NE reformation, so it is unable to distinguish between anaphase and early telophase. Anaphase ends about 5 minutes after anaphase onset, and telophase begins. The time-lapse data in this paper does not allow for strict distinction between mitotic phases, and as a result, it calls it anaphase based on the chromosome morphology, but the period 5 minutes before the anaphase onset should be called anaphase, and the period 5 minutes after should be called telophase (or early telophase).
2. line 81, "--- in the process (Fig. 1C, D)": Figure 1C is not appropriate to cite here. Wouldn't E and F be more appropriate? Fig. 1E and 1F are not cited in this paper. Please cite C, E, and F in the appropriate places.
3. line 110: Fig. 2D seems missing in the text. please cite the figure in the appropriate place.
4. line 112, "after anaphase onset": This expression is not incorrect, but it would be better to write "during telophase"
5. line 126, 152, 167, 687, and sup figure legends "Life cell imaging": The same term is used in many places. Another term is live cell imaging. Wouldn't "live cell imaging" be more appropriate?
6. lines 352-356, 4 mg/ml RNA stock solution: Is the final concentration of the RNA solution 4 mg/ml correct? If so, double-check the amounts of each reagent.
7. line 406, 409, CO2: Correct to CO₂

8. line 748: Color figures (Fig. 4A-D) require a description of the colors. Add a description.

9. In the methods section, the phrase "by an expert biologist" appears many times, but I did not clearly understand what it meant. It should be made clear what kind of abilities and knowledge a biologist with is required. Or, it should be stated what software was used to examine the method.

10. In Sup Fig. 1, there are some figures where the deviation is taken as sem and some where it is taken as sd. Please check if the description is correct.

11. In the legend of Sup Fig. S3, the figure is A-E, but the legends are A-G. Correct it. In the legend of Sup Fig. S3D, it says "the means", but "+/- sd" may be necessary. Correct it if necessary.

12. The left bar in Sup Fig. S4E is wider. Is this OK?

13. The legend of Sup Fig. S4C says "Ki-67 A", but isn't it "Ki-67"? Please correct if necessary.

14. In Sup Fig. S5, I could not understand why the results of control EFGP were so different between experiments.

15. Although the legend of Sup Fig. S6B says "time-lapse images," actually only a single image is shown for each experiment. Please fix the legend or image.

16. Please describe what colors are in Supplementary Figure S8.

Version 1:

Reviewer comments:

Reviewer #1

(Remarks to the Author)

The authors have fully addressed my concerns. I recommend the publication of this paper.

Reviewer #2

(Remarks to the Author)

The authors have satisfactorily answered my concerns.

Reviewer #3

(Remarks to the Author)

The authors have addressed all of my concerns. I have no further concerns.

Response to reviewer comments

Thank you for reviewing this manuscript and for your invaluable comments. We have addressed all the points raised by the reviewers, as detailed below. Although not requested, we have included siRNA experiments with a siRNA-resistant version of eIF4A1, which are now presented in Figures 2G/H. We believe this addition adds value to the manuscript.

We have replaced the RNA staining method using Syto-RNA with an alternative approach that employs click chemistry to label RNA on mitotic chromosomes. Given that Syto-RNA may also stain DNA (as noted in sources such as <https://www.aatbio.com/resources/application-notes/selective-analysis-of-rna-in-live-and-fixed-cells-with-strandbrite-rna-green> and <https://onlinelibrary.wiley.com/doi/full/10.1002/cyto.10076>), we cannot rule out the possibility that the stronger signal observed upon downregulation of eIF4A1/2 is due to increased chromatin compaction. Therefore, we adopted a protocol recently utilized by Hernandez-Armendariz et al. (Mol Cell 84(17):3254-3270.e9; doi: 10.1016/j.molcel.2024.07.022) for labeling RNA on mitotic chromosomes via click chemistry. Interestingly, our findings indicate that downregulation of eIF4A1/2 leads to a reduction in perichromatin RNA signal, which is now included in Figures 3A/B, replacing the previous Syto RNA staining.

Additionally, we have provided data demonstrating that eIF4A1 does not enrich or colocalize with DDX18/21/27-RNA condensates (Fig. 7B). Based on these new insights and additional novel data requested by the reviewers (see below), we have revised our model to propose that eIF4A1/2 functions during mitosis as a cytosolic RNA chaperone that enhances RNA mobility, which is essential for efficient chromatin decondensation.

Point to point response:

Reviewer #1

Jühlen, Wiesmann et al. demonstrate that eIF4A1/2, known as translation factors, facilitate chromosome decondensation in late mitosis through their RNA binding activity. They propose that eIF4A1/2 induce the dissociation of RNA from perichromatin by disrupting RNA-protein condensates.

This paper will be of great interest to researchers in the field, and I find it exciting due to its significance in advancing our understanding of chromosome decondensation during the transition from mitosis to interphase. However, several issues need to be addressed to ensure that the claims and the model are adequately supported by the data.

Our answer: Thanks for the overall positive evaluation. We have addressed the points raised to support the model as specified below.

Major point

There is not enough data to support the proposed model

The authors argue that protein-RNA condensates are dissociated from perichromatin by eIF4A1/2 (Fig. 6G). However, some inconsistencies need to be addressed, and additional data are required to support/modify this model.

Firstly, while the authors clearly show that RNA-protein (DDX18, DDX21, or DDX27) condensates are disrupted by the addition of eIF4A1/2/3 (Fig. 6A-D), they have not checked for the dissociation of these proteins from perichromatin. It is expected that dissociation would be observed with the add-back of eIF4A1/2 and retention with depletion. Confusingly, with eIF4A1/2 knockdown, CCDC137, used as a perichromatin marker, formed clusters and moved away from the mitotic chromosome, and the signal for KI67, the central organizer of perichromatin, also weakened (Fig. 3C and H). How can this be explained?

Our answer: We have now included immunofluorescence data showing that the DDX18 and DDX27 signals are reduced at mitotic chromosomes upon eIF4A1/2 downregulation, consistent with the reduced CCDC137 and Ki-67 signal. These data are now included as Fig. 3. As for CCDC137 we see an increased cytosolic mislocalization of DDX18 and DDX27 upon eIF4A1/2 downregulation (Fig. 3E,F,I J). Likely, the condensate experiment reflects the cytosolic behavior of RNA condensates. We have adapted the discussion accordingly.

Secondly, the authors demonstrate that eIF4A1/2 knockdown leads to the retention of RNA on perichromatin (Fig. 3B). However, to support the proposed model, the authors need to test whether eIF4A1/2 actively dissociates RNA from perichromatin. Employing the rapamycin system shown in Fig. 5 would be useful. Additionally, when quantifying RNA at perichromatin, the authors need to measure the signal of RNA co-localized with EGFP-CCDC/KI67. Otherwise, the results are not conclusive.

Our answer: As we cannot exclude that the Syto-RNA also stains DNA (see also above) we cannot exclude that the stronger signal observed upon eIF4A1/2 downregulation is due to higher chromatin compaction. We therefore labeled RNA on mitotic chromosomes using click-chemistry following a protocol recently employed by Hernandez-Armendariz et al (Mol Cell 84(17):3254-3270.e9. doi: 10.1016/j.molcel.2024.07.022). Surprisingly, we find that eIF4A1/2 downregulation reduces the perichromatin RNA signal. These data are now included as Figure 3A/B and replace the Syto RNA staining. We tried to do the reverse experiment attempting eIF4A1 tethering on the chromatin. As the RNA staining relies on Alexa-488 labeling we had to use eBFP-eIF4A1-FRB instead of eGFP-eIF4A1-FRB. We see a slight increase in RNA signal upon eIF4A1 chromatin tethering. As the number of cells we could analyze, and the RNA signal is generally low the quantitation is less robust and we prefer not to include these data in the manuscript. As the EGFP-CCDC137/EGFP-Ki-67 label overlaps with the RNA staining (Alexa 488) we cannot image both signals concomitantly.

Quantitation of the RNA perichromatin signal (EU-labelled RNA is visualized with click-chemistry via Alex 488-label) in fixed H2B-mPum-FKBP HeLa cells expressing EBFP-FRB-eIF4A1 at different mitotic phases. eIF4A1 chromatin tethering has been induced with 200 nM rapamycin

Lastly, it seems challenging to demonstrate that the findings from the condensate assay are also occurring at perichromatin. The essence of the decondensation mechanism proposed by the authors appears to lie in the RNA-binding affinity of eIF4A1/2. An additional experiment to consider would be to investigate whether eIF4A1/2 has a stronger binding affinity for RNA compared to other RNA-binding proteins localized to perichromatin. If eIF4A1/2 can competitively exclude RNA from these proteins, it may be possible to propose a model more consistent with the data and suggest a link to phase separation mechanisms by referring to previous studies. This may temper the authors' claims, but the significance of the findings remains substantial.

Our answer: In the light of the reduced RNA-staining upon eIF4A1/2 downregulation we consider the action of eIF4A1/2 in mitosis as cytoplasmic to enhance RNA mobility. Also, in this context it is interesting to consider the affinity of eIF4A1/2 towards RNA as compared to other RNA binding factors. This is now included in the discussion – thanks for pointing at this.

Minor points

1. Title suggestion: Should it be changed to eIF4A1/2 instead of eIF4A1?

Our answer: This is a very valid suggestion. We have changed the title and adapted the abstract accordingly.

2. Some figures are too small, especially Fig3A, C, D, and G. The chromosome/protein localization indicated by the arrowheads cannot be identified.

Our answer: We have split the original Figure 3 in two (Fig. 3 and Fig. 4) adapted the figures accordingly.

3. Please provide the author's explanation for the prolonged telophase observed with eIF4H knockdown.

Our answer: For eIF4H knockdown we do not see a clear decondensation defect (measured by the increase in chromatin area, Fig. S1G). Telophase requires many processes like nuclear envelope reformation, spindle disassembly or nuclear import re-initiation unrelated to chromatin decondensation and we cannot exclude that eIF4H contributes to one of these. Of note, eIF4H chromatin tethering (Fig. S7) does not impact chromatin decondensation. We have commented these points in the discussion.

4. What type of RNA is used in the condensate assay (Fig6 A-D)? Please specify in the methods section.

Our answer: We use yeast total RNA (Roche, 10109223001). The procedure is described in detail in the method section (line 420ff).

Reviewer #2:

Jühlen et al. present data in support of their hypothesis that eIF4A1 plays a role in driving de-condensation of chromatin during mitotic exit. Using *Xenopus* chromatin clusters they demonstrate that recombinant eIF4A can promote decondensation and build on this to demonstrate that RNA binding, but not helicase activity is necessary for this activity. They conclude that eIF4A is functioning as an RNA chaperone rather than a helicase to promote this activity. Overall all, the data is of high quality and largely supports their conclusions. I have a couple comments that should be addressed before publication with point #5 being the most relevant.

Our answer: Thanks for the very positive evaluation. We have addressed the points raised as specified below.

1. Figure 1: I realize these are in vitro assays in which recombinant proteins were added. However, it would be nice to have some loading control for these experiments. Particularly in light of the depleted lysate in which there is no band present.

Our answer: Thanks for pointing at this. Nup62, an abundant nuclear pore complex component has now been included as “loading control” in Fig. 1C, F.

2. Figure 2: Again, a minor point, but demonstrating depletion efficiency for the experiment in figures 2A – C is necessary. Is it the same as shown in 2D? If so, this should be presented 1st. If not, the data for this experiment should be shown.

Our answer: Fig. 2D corresponds to Fig. 2A-C. We did not mention Fig. 2D in the original manuscript, but this mistake is now corrected.

3. The authors keep referring to eIF4F, but only every analyze eIF4G and eIF4A. These experiments should be conducted with eIF4E to formalize their statement that eIF4F is not necessary. Alternatively, the wording should be altered.

Our answer: eIF4E is included in the siRNA analysis (Fig. S1F/G) but not in the tethering assays because it is not a directly interact of eiF4A1/2. Where required we changed the wording in the manuscript (line 181, 204).

4. The tethering experiments are intriguing. However, I have some concerns about the use of rapamycin in this experiment as eIF4A activity is inhibited by rapamycin. Granted, this is thought to be through eIF4B phosphorylation and is likely not involved given the authors have demonstrated that the effect they see is independent of helicase activity or eIF4B. However, this potential complication should be discussed.

Our answer: Thanks for pointing at this. We have included this in the discussion (line 339ff). Most importantly, rapamycin addition to cells to induce chromatin tethering of other eIF4F components does not enhance chromatin decondensation arguing against an inhibitory effect of rapamycin in this context.

5. The ability of eIF4A to resolve helicase granules is interesting, however, I'm not convinced what this is showing in terms of decondensation of chromatin. Is it previously established that RNA:helicase complexes are involved in maintaining the perichromatin? I understand that helicases are implicated in LLPS resolution, but is there any connection between these helicases and perichromatin? If so, this could be explained better. As currently presented, it's an interesting observation, but unclear as to how it relates. Further, this assay is showing that eIF4A3 is able to dissolve these condensates. This is in contrast to previous data demonstrating that eIF4A3 is not involved in decondensation. This further lessens the impact of this figure.

Our answer: In the light of the reduced RNA-staining upon eIF4A1/2 downregulation we consider the action of eIF4A1/2 in mitosis as cytoplasmic to enhance RNA mobility. We have adapted the discussion accordingly. Of note, expression levels of eIF4A3 are considerably lower than those of eIF4A1/2 ($3,28 \cdot 10^5$ copies per U2OS cell for eIF4A3 versus $2,22 \cdot 10^6$ for eIF4A1 and $2,91 \cdot 10^5$ for eIF4A2, Beck et al, 2011 (<https://www.embopress.org/doi/full/10.1038/msb.2011.82>)). An eIF4A3 effect might be only detectable upon overexpression or if tethered to chromatin – we have commented this in the discussion (line 321ff). In addition, even at equimolar ratios eIF4A3 seems to be less efficient than eIF4A1 or eIF4A2 (see Fig. 7).

Reviewer #3

The process of chromosome decondensation during telophase of mitosis is important for eukaryotes undergoing open mitosis to establish interphase chromatin structure. However, little is known about the molecules involved in this process. This paper discovered eIF4A1 as a protein involved in this process. eIF4A1 is one of the components of the eIF4F complex that works in the initiation of translation, and has ATPase activity and RNA helicase activity when working in translation initiation. However, this paper clarified using the cell-free system, in vivo (in cell) experiments, and in vitro experiments that these activities are not required for chromosome decondensation activity, and that RNA binding ability plays a key role instead. This paper is excellent in that it has discovered a previously unknown factor involved in decondensation. Moreover, it is highly novel that a factor involved in the initiation of protein synthesis has such a function completely independent of its activity. The experimental methodology is sound, and I mostly agree with the conclusions drawn from the results.

I have a few comments that should be revised before publication.

Our answer: Thanks for the very positive evaluation. We have addressed the points raised as specified below.

Minor comments

1. line 56, "during anaphase and early telophase": I recommend removing the word "anaphase" from this sentence. The reason is that chromosome decondensation occurs with nuclear envelope (NE) reformation, but in anaphase, NE reformation does not occur, so there is no need to include anaphase. This paper does not observe the NE reformation, so it is unable to distinguish between anaphase and early telophase. Anaphase ends about 5 minutes after anaphase onset, and telophase begins. The time-lapse data in this

paper does not allow for strict distinction between mitotic phases, and as a result, it calls it anaphase based on the chromosome morphology, but the period 5 minutes before the anaphase onset should be called anaphase, and the period 5 minutes after should be called telophase (or early telophase).

Our answer: We have changed the sentence to “critically enabling chromatin decondensation during telophase.”

2. line 81, "--- in the process (Fig. 1C, D)": Figure 1C is not appropriate to cite here. Wouldn't E and F be more appropriate? Fig. 1E and 1F are not cited in this paper. Please cite C, E, and F in the appropriate places.

Our answer: Fig. 1C is now cited and Fig. 1 D, E, F correctly mentioned in the text.

3. line 110: Fig. 2D seems missing in the text. please cite the figure in the appropriate place.

Our answer: The mistake has been corrected.

4. line 112, "after anaphase onset": This expression is not incorrect, but it would be better to write "during telophase"

Our answer: "after anaphase onset" has been changed to “during telophase”

5. line 126, 152, 167, 687, and sup figure legends "Life cell imaging": The same term is used in many places. Another term is live cell imaging. Wouldn't "live cell imaging" be more appropriate?

Our answer: Has been corrected throughout the text

6. lines 352-356, 4 mg/ml RNA stock solution: Is the final concentration of the RNA solution 4 mg/ml correct? If so, double-check the amounts of each reagent.

Our answer: The final concentration of RNA in the in vitro phase separation experiments (now Fig. 7) is 1 mg/ml as clarified in the method section and noted in the figure legend.

7. line 406, 409, CO₂: Correct to CO₂

Our answer. Has been corrected in line 406. The environmental control system UNO-T-H-CO₂ is written without subscripted character.

8. line 748: Color figures (Fig. 4A-D) require a description of the colors. Add a description.

Our answer: Thanks, this has been included in the figure legend (now Fig. 5).

9. In the methods section, the phrase "by an expert biologist" appears many times, but I did not clearly understand what it meant. It should be made clear what kind of abilities and knowledge a biologist with is required. Or, it should be stated what software was used to examine the method.

Our answer: We agree this is too undefined. We have removed this phrase and modified the method to be clearer.

10. In Sup Fig. 1, there are some figures where the deviation is taken as sem and some where it is taken as sd. Please check if the description is correct.

Our answer: We show the SEM if data from more than one experiment are pooled to show how far the sample mean is likely to be off from the real mean. For data generated in one single experiment we show the SD to indicate how far the measurements differ from the sample mean.

11. In the legend of Sup Fig. S3, the figure is A-E, but the legends are A-G. Correct it. In the legend of Sup Fig. S3D, it says "the means", but "+/- sd" may be necessary. Correct it if necessary.

Our answer: The figure legend has been corrected. As for panel D data from three independent experiments were pooled we show here the SEMs.

12. The left bar in Sup Fig. S4E is wider. Is this OK?

Our answer: Thanks, this has been corrected

13. The legend of Sup Fig. S4C says "Ki-67 A", but isn't it "Ki-67"? Please correct if necessary.

Our answer: Has been corrected.

14. In Sup Fig. S5, I could not understand why the results of control EFGP were so different between experiments.

Our answer: The data are based on two sets of experiments. The first with a frame rate 3 min (A, B, C, E), the second with 2 min (D, F). Accordingly, the control data could not be pooled. The same applies to panels G-L: G, H, I and K are from the same experiment, J and L from a second.

15. Although the legend of Sup Fig. S6B says "time-lapse images," actually only a single image is shown for each experiment. Please fix the legend or image.

Our answer: Has been corrected

16. Please describe what colors are in Supplementary Figure S8.

Our answer: Thanks, this has now been included in the figure legend.